# Towards Safe and Generalizable Treatment Strategies in Healthcare via RL and PAC-Bayesian Computations

## Abstract

Reinforcement learning (RL) offers a promising paradigm for optimizing treatment strategies that adapt over time to patient responses. However, the deployment of RL in clinical settings is hindered by the lack of generalization guarantees, an especially critical concern given the high-stakes nature of this domain. Existing generalization bounds for sequence data are either vacuous or rely on relaxations of the independence condition, which often produce non-sharp bounds and limit their applicability to RL. In this work, we derive a novel PAC-Bayesian generalization bound for RL that explicitly accounts for temporal dependencies arising from Markovian data. Our key technical contribution integrates a bounded-differences condition on the negative empirical return to establish the applicability of a McDiarmid-style concentration inequality tailored to dependent sequences such as Markov Decision Processes. This leads to a PAC-Bayes bound with explicit dependence on the Markov chain's mixing time. We show that our bound can be directly applied to off-policy RL algorithms in continuous control settings, such as Soft Actor-Critic. Empirically, we demonstrate that our bound yields meaningful confidence certificates for treatment policies in simulated healthcare environments, providing high-probability guarantees on policy performance. Our framework equips practitioners with a tool to assess whether an RL-based intervention meets predefined safety thresholds. Furthermore, by closing the gap between learning theory and clinical applicability, this work advances the development of reliable RL systems for sensitive domains such as personalized healthcare.

## 1 Introduction

Reinforcement learning (RL) is increasingly being explored for high-stakes decisions in healthcare, where the promise is to tailor treatments to individual patients *based on treatment history* and improve outcomes over time [1, 2]. Unlike traditional static models, RL agents are capable of learning from the clinician's observation–action cycle: observe a patient's state (e.g., vital signs, symptom scores), select an intervention (e.g., medication adjustment, therapy session), then observe the outcome and update the model accordingly. This sequential framework has spurred applications ranging from critical care management of sepsis [2, 3] to precision drug dosing [3].

The RL paradigm applied to healthcare offers a principled framework for optimizing sequential decisions based on patient responses. For instance, a recent study [1] introduced the notion of *medical dead-ends*, meaning critical states from which all future trajectories lead to adverse outcomes, and utilized RL to proactively recognize treatment paths associated with these dangerous declines. These kinds of applications illustrate the potential of RL to enhance decision-making in safety-critical domains such as medical care. However, the medical setting also underscores the paramount need for reliability–an RL policy's recommendations can literally be life-saving or life-threatening.

Submitted to 39th Conference on Neural Information Processing Systems (NeurIPS 2025). Do not distribute.

There is a widely unmet demand for rigorous, high-confidence guarantees on the generalization capabilities of machine learning models [4], particularly RL for healthcare to ensure that algorithm performance on patients' data meets acceptable standards. Without formal generalization assurances, clinicians might rightly question the reliability and robustness of RL-derived treatment recommendations for unseen patient populations, limiting broader adoption in practice.

A core challenge in providing guarantees that could drive trust in RL for healthcare applications is the *sequential, dependent nature* of RL data. Most standard generalization bounds [5] and concentration inequalities [6, 7] rely on the assumption that samples are independent and identically distributed (i.i.d.). However, this assumption is violated when learning from trajectories generated by a Markov decision process. In this work, we address this challenge using the PAC-Bayesian framework [8, 9, 10, 11], which yields data-dependent generalization guarantees that are often tighter, naturally incorporate prior knowledge, and are straightforward to optimize. By balancing empirical risk against model complexity via a prior–posterior divergence, PAC-Bayes offers a principled way to quantify uncertainty and reason about generalization [4].

Previous efforts to bring PAC-Bayes to RL include [12, 13], who derived bounds for batch RL with implicit constants that dependent on the mixing-time which can limit practical utility, and more recently [14], who extended the analysis to the context of actor-critic learning to encourage exploration. While conceptually exciting, the bounds in these previous works remained largely vacuous, reflecting a focus on learning-theoretic algorithm development rather than on deriving tight performance guarantees. These results of these previous works thus underscore the promise of PAC-Bayesian analysis for RL but also highlight the need for bounds that better capture the dependency structure of clinical trajectories: existing results either rely on stringent mixing assumptions or yield overly loose guarantees, leaving a significant gap in providing tight confidence certificates for RL algorithms.

In this work, we bridge this gap by deriving a *novel PAC-Bayesian generalization bound for reinforcement learning that explicitly handles Markov dependencies* in the data. The core technical contribution is the integration of concentration inequalities suited for dependent sequences into the PAC-Bayes analysis. In particular, we leverage the Marton coupling and Markov chain partitioning [15] to establish a *McDiarmid-type bounded differences inequality for Markov chains*. Our resulting PAC-Bayesian bound retains an explicit dependence on mixing time, thus preserving the interpretability and theoretical grounding of classical approaches while achieving *tighter constant factors* that render the bound non-vacuous for realistic trajectory lengths.

Beyond its theoretical contribution, our bound has direct practical implications for safe and reliable RL in healthcare. By providing a high-probability certificate on a policy's true return, practitioners can assess, before deployment, whether an RL-based treatment strategy meets predefined safety and efficacy standards. For example, in adaptive dosing for chronic conditions, our bound can guarantee with high confidence that the expected patient health score will not fall below a critical threshold. Looking forward, we envision applying this framework to mental health interventions, where data scarcity and patient vulnerability amplify the need for trustworthy RL policies [16].

In summary, by integrating mixing-time explicit bounds with advanced coupling methods, we deliver a PAC-Bayesian guarantee that is *rigorously grounded* and we demonstrate in experiments that it is *practically meaningful*, paving the way for reinforcement learning algorithms that are provably safe and effective in real-world healthcare settings.

The remainder of this paper is organized as follows. Section 2 reviews necessary preliminaries—reinforcement learning, PAC-Bayesian learning, and related work—and summarizes our contributions. In Section 3 we develop the theoretical core by deriving a new PAC-Bayesian generalization bound for RL that explicitly accounts for Markovian dependence via a McDiarmid-type concentration inequality. Section 3.1 introduces PB-SAC, a practical actor–critic algorithm that operationalizes this bound. Section 4 describes our experimental setup on an ICU-Sepsis simulator and standard continuous-control benchmarks, and presents empirical results demonstrating that PB-SAC delivers meaningful confidence certificates without sacrificing performance. Finally, Section 5 concludes with a discussion of implications, limitations, and directions for future work in PAC-Bayesian reinforcement learning.

## 2  Preliminaries

We briefly recall the reinforcement learning and statistical learning theory concepts we rely on throughout the paper. The exposition is intentionally concise—the goal is to fix notation and state the learning theory principles that underpin our results. We adpot the notation from [17].

### 2.1  Reinforcement Learning & Related Works

Reinforcement Learning (RL) studies how an *agent* learns to make sequential decisions through interaction with an environment. Formally, the environment is modeled as a (possibly unknown) Markov Decision Process (MDP) $\mathcal{M} = (\mathcal{S}, \mathcal{A}, \mathbb{P}, R, \gamma)$, where $\mathcal{S}$ is the state space, $\mathcal{A}$ the action space, $\mathbb{P}(s' \mid s, a)$ the transition kernel, $R(s, a)$ the reward function bounded in $[0, R_{\max}]$, and $\gamma \in (0, 1)$ the discount factor. At each time $t$, the agent observes a state $S_t \in \mathcal{S}$, chooses an action $A_t \in \mathcal{A}$ according to a *policy* $\pi(a|s)$, receives a reward $R_{t+1} = R(S_t, A_t)$, and transitions to $S_{t+1} \sim \mathbb{P}(\cdot \mid S_t, A_t)$.

The agent's objective is to maximise the *expected discounted return*

$$G_t = \sum_{k=0}^{\infty} \gamma^k R_{t+k+1}, \qquad V_\pi(s) = \mathbb{E}_{\pi, \mathbb{P}}\big[G_t \mid S_t = s\big], \tag{1}$$

where $V_\pi$ is the state–value function. The optimal value function $V^\star(s) = \sup_\pi V_\pi(s)$ satisfies the Bellman optimality equation

$$V^\star(s) = \max_{a \in \mathcal{A}} \Big\{ R(s, a) + \gamma\, \mathbb{E}_{s' \sim \mathbb{P}}\big[V^\star(s') \mid s, a\big] \Big\}. \tag{2}$$

RL algorithms learn either directly a policy (policy–gradient and actor–critic methods [18, 19, 20]) or an action–value function $Q_\pi(s, a)$ (value–based methods such as Q-learning and its deep variants [21]). Model–free approaches dispense with an explicit model of $\mathbb{P}$, while model–based methods leverage or learn a transition model to plan.

**PAC-Bayesian analysis for sequential data.**  Classical PAC-Bayesian theory [8, 22, 9, 23, 24, 10, 17, 11] assumes i.i.d. samples, but several authors have extended it to dependent data. Ralaivola *et al.* introduced a *chromatic PAC-Bayes* bound for $\beta$-mixing sequences, recasting temporal dependence as a graph–coloring problem that preserves a PAC-style risk certificate [25]. Seldin *et al.* pioneered a martingale-based PAC-Bayes approach, showing how to integrate concentration inequalities for dependent observations (e.g. Hoeffding-Azuma for martingales) with PAC-Bayes bounds [26, 27]. Further generalizations to weakly dependent series were obtained by Alquier & Wintenburger via oracle inequalities for time-series forecasting [28]. These contributions established that PAC-Bayes remains applicable when observations are correlated, provided one can quantify the dependence.

**PAC-Bayes in reinforcement learning.**  Early applications to RL include Fard *et al.*, who derived a batch RL bound relying on Samson's inequality for uniformly ergodic Markov chains [12, 13]. They demonstrated empirically that PAC-Bayesian model selection can indeed improve policy value estimation by taking the prior when it is informative and discarding it when missleading. Although insightful, the constants scale poorly with the horizon, often making the bound vacuous in practice. More recently, Tasdighi *et al.* embedded a PAC-Bayesian critic ensemble inside an actor–critic algorithm to encourage deep exploration, but did not compute a certified return gap [14]. Zhang *et al.* used task-adaptive PAC-Bayes priors for lifelong RL [29]. Despite these advances, prior work has not produced a sharp PAC-Bayes bound that is simultaneously translated into non-vacuous and tight certificates for modern off-policy methods such as Soft Actor-Critic, and depends on explicit constants making it easy to compute in practice.

**Our contribution in context.**  We close this gap by deriving a PAC-Bayesian value-error bound whose leading constant is proportional to the Markov chain's mixing time. Compared with previous PAC-Bayes works, our result (i) obtains tighter scaling for discount factors typical in RL. (ii) embed the new bound in a Soft Actor–Critic framework and show empirically that the resulting *PB-SAC* algorithm can monitor and minimize its certified return gap throughout training. To our knowledge, this is the first demonstration that PAC-Bayesian guarantees with explicit temporal–dependence

constants can inform the hyper-parameter choices of deep off-policy RL while remaining non-vacuous in realistic continuous-control domains.

## 2.2 PAC-Bayes Learning

Let $(\mathcal{X}, \mathcal{Y}, \mathcal{D})$ be a supervised learning task, the domain is taken to be the product $\mathcal{Z} = \mathcal{X} \times \mathcal{Y}$, where $\mathcal{X} \subseteq \mathbb{R}^d$ is the feature space and $\mathcal{Y}$ the label space ($\mathcal{Y} \subseteq N$ for classification problems, or $\mathcal{Y} \subseteq \mathbb{R}$ for regression ones). We assume an unknown data distribution $\mathcal{D}$ over $\mathcal{Z}$, with $\mathcal{D}_{\mathcal{X}}$ denoting the marginal distribution on $\mathcal{X}$. We observe a training sample $S = \{(\boldsymbol{x}_i, y_i)\}_{i=1}^m$, where each pair $(\boldsymbol{x}_i, y_i) \in \mathcal{Z}$ is drawn independently and identically distributed (i.i.d.) from $\mathcal{D}$, that is, $S \sim \mathcal{D}^m$. This sample is provided to the learning algorithm. Given a sample $S$, the learning algorithm returns a measurable prediction function $f_\theta : \mathcal{X} \to \mathcal{Y}$, also referred to as a hypothesis, parametrized by $\theta \in \Theta$, where $\Theta$ denotes the set of all admissible parameter vectors (i.e., the hypothesis class). The "quality" of a hypothesis $f_\theta$ is typically assessed through a measurable loss function $\ell : \mathcal{Y} \times \mathcal{Y} \to \mathbb{R}_+$, which quantifies the discrepancy between predicted and true outputs. The performance of a hypothesis is measured by its *true risk*, and its *empirical risk* on the training sample $S$,

$$\mathcal{L}(\boldsymbol{\theta}) = \mathop{\mathbb{E}}_{(x,y)\sim\mathcal{D}} \big[\ell\big(f_\theta(x), y\big)\big], \quad \hat{\mathcal{L}}_S(\boldsymbol{\theta}) = \frac{1}{m}\sum_{i=1}^m \ell\big(f_\theta(x_i), y_i\big),$$

In supervised machine learning, the goal is to learn a hypothesis $f_\theta$ that accurately predicts a label $y \in \mathcal{Y}$ for a new input $x \in \mathcal{X}$, based on a training dataset $S = \{(x_i, y_i)\}_{i=1}^m$. A central question is: how can we ensure that the learned function $f_\theta$ will perform well on unseen data?

$$\Pr_{S\sim\mathcal{D}^m}\big\{\mathcal{L}(\boldsymbol{\theta}) \leq \hat{\mathcal{L}}_S(\boldsymbol{\theta}) + \epsilon\big\} \ \geq \ 1 - \delta.$$

Concrete PAC bounds specify how large $m$ must be (or how large the gap $\epsilon$ can be) in terms of properties of the hypothesis class—e.g. VC-dimension, Rademacher complexity, stability, compression, etc. All of those treat $f_\theta$ as a deterministic output of the algorithm.

The **PAC-Bayesian** framework [8, 9, 10, 17, 11] extends the PAC learning paradigm to analyze the generalization performance of stochastic learning algorithms. Instead of selecting a single hypothesis, this approach considers a distribution over a set of candidate models. Let $\Theta$ denote the set of parameters defining a family of prediction functions $\{f_\theta : \mathcal{X} \to \mathcal{Y}\}_{\theta\in\Theta}$. Prior to observing data, a *prior* distribution $\mu \in \mathcal{P}(\Theta)$ is specified over $\Theta$. Upon receiving a training sample $S \sim \mathcal{D}^m$, the learning algorithm selects a *posterior* distribution $\rho \in \mathcal{P}(\Theta)$, potentially dependent on $S$. PAC-Bayesian theory provides high-probability bounds on the population Gibbs risk $\mathbb{E}_{f_\theta\sim\rho}[\mathcal{L}(\boldsymbol{\theta})]$ in terms of the empirical Gibbs risk $\mathbb{E}_{f_\theta\sim\rho}[\hat{\mathcal{L}}_S(\boldsymbol{\theta})]$ and an additional term that measures the dependence of the posterior distribution $\rho$. This additional term involves an information measure—typically the Kullback-Leibler divergence $\mathrm{KL}(\rho\|\mu)$—between the data-dependent posterior $\rho \in \mathcal{P}(\Theta)$ and a prior $\mu \in \mathcal{P}(\Theta)$, chosen independently of the data. Formally, for any $\lambda > 0$ and with probability at least $1 - \delta$ over the choice of the training sample $S$, the following inequality holds:

$$\mathop{\mathbb{E}}_{f_\theta\sim\rho}[\mathcal{L}(\boldsymbol{\theta})] \ \leq \ \mathop{\mathbb{E}}_{f_\theta\sim\rho}[\hat{\mathcal{L}}_S(\boldsymbol{\theta})] \ + \ \frac{1}{\lambda}\Big(\mathrm{KL}(\rho\|\mu) + \ln\frac{1}{\delta} + \Psi_{\ell,\mu}(\lambda, n)\Big) \tag{3}$$

$$\Psi_{\ell,\mu}(\lambda, m) = \ln \mathop{\mathbb{E}}_{f_\theta\sim\mu}\Big[\exp\big(\lambda\big(\mathcal{L}(\boldsymbol{\theta}) - \hat{\mathcal{L}}_S(\boldsymbol{\theta})\big)\big)\Big]$$

Compared with classical PAC guarantees, PAC-Bayes offers two advantages that are critical for reinforcement learning; *Data-dependent priors [30]*–when $\mu$ can itself depend on previous data (*e.g.* earlier tasks or behavioural trajectories) [29], the bound adapts to the knowledge already acquired, tightening $\mathrm{KL}(\rho\|\mu)$; *Fine-grained control via* $\Psi$–by tailoring the concentration inequality used to upper-bound $\Psi$ one can incorporate dependence structures such as martingales [27, 26], $\beta$-mixing [25, 31] sequences or Markov chains [13, 14]—exactly the scenario in which RL trajectories are collected.

## 3 PAC-Bayes Framework for RL

As outlined earlier, our objective is to establish a *high-probability* PAC-Bayes **value-error** bound for a policy operating in a Markov decision process (MDP) when the training data are *dependent* trajectories—possibly gathered under an off-policy behaviour strategy. In this section, we begin by fixing notation, then present the main results; all proofs are deferred to Appendix B.

Let $\mathcal{M} = (\mathcal{S}, \mathcal{A}, \mathbb{P}, R, \gamma)$ be a discounted Markov Decision Process (MDP), where $\mathcal{S}$ and $\mathcal{A}$ are the state and action spaces, $\mathcal{P}$ is the transition kernel, $R$ is the reward function such that $R_t \in [0, R_{\max}]$, and $\gamma \in (0, 1)$ is the discount factor. A policy $\pi_\theta$ induces a (not necessarily *time-homogeneous*) Markov chain $\xi = (S_1, A_1, R_1, S_2, \ldots, S_H) \sim \nu, \mathbb{P}, \pi_\theta, R$, where $\nu$ denotes the initial state distribution and $H \leq \infty$ is the trajectory horizon (finite or infinite). Our analysis naturally extends to the infinite-horizon case ($H = \infty$).

We assume access to a dataset $\mathfrak{D} = \{\xi^{(1)}, \ldots, \xi^{(T)}\}$ of $T$ trajectories (*i.e.,* $N = HT$ transitions in total), collected using a behavior policy $\pi_\theta$, parameterized by $\theta \in \Theta$. The parameters $\theta$ are drawn from a distribution $\rho \in \mathcal{P}(\Theta)$, where $\Pi = \{\pi_\theta : \theta \in \Theta\}$ denotes the policy class. Henceforth, we write $\xi \sim \mathcal{M}$ (**resp.** $\mathfrak{D} \sim \mathcal{M}^{(T)}$) to denote sampling a trajectory (**resp.** a set $\mathfrak{D}$ of $T$ trajectories) under the environment dynamics $\mathbb{P}$, initial state distribution $\nu$, policy $\pi_\theta$, and reward function $R$, in order to avoid notational overload.

We define the discounted return of a trajectory and its expected value under policy $\pi_\theta$ as:

$$G(\xi) = \sum_{k=0}^{H-1} \gamma^k R_{k+1} \quad \text{and} \quad V_{\pi_\theta} = \mathbb{E}_{\xi \sim \mathcal{M}}[G(\xi)]. \tag{4}$$

We now define the expected (true) loss and its empirical counterpart:

$$\mathcal{L}(\theta) = \begin{cases} - \mathbb{E}_{\xi \sim \mathcal{M}}[G(\xi)] \\ = \mathbb{E}_{\mathfrak{D} \sim \mathcal{M}^{(T)}}[\hat{\mathcal{L}}_{\mathfrak{D}}(\theta)] \end{cases} \quad \text{where} \quad \hat{\mathcal{L}}_{\mathfrak{D}}(\theta) = -\frac{1}{T} \sum_{j=1}^{T} G(\xi^{(j)}) \tag{5}$$

**Prior and posterior over policies.** Following the PAC-Bayesian paradigm we endow $\Theta$ with a *prior* distribution $\mu \in \mathcal{P}(\Theta)$, selected independently of the data, and a *posterior* distribution $\rho \in \mathcal{P}(\Theta)$, chosen after observing the sample $\mathfrak{D}$. This PAC-Bayesian formalism allows us to reason about the generalization properties of randomized policies drawn from $\rho$, with theoretical guarantees based on their divergence from the prior $\mu$.

**A bounded-differences property for the empirical loss.** The following lemma shows that *changing one transition* in the data results in quantitative bounded effect of the empirical loss defined in (5).

**Lemma 3.1 (Bounded differences)** *Let $\mathfrak{D}$ be a set of trajectories and $\theta \in \Theta$ be fixed policy parameters. Suppose we form $\bar{\mathfrak{D}}$ by changing one transition, say the transition at time step $h \in [H]$ of trajectory $j \in [T]$, where $\xi_h^{(j)} = (s, a, r, s')$ is replaced with $\bar{\xi}_h^{(j)} = (\bar{s}, \bar{a}, \bar{r}, \bar{s}')$. Then, there exists $c \in \mathbb{R}_+^{H \times T}$ such that*

$$\left| \hat{\mathcal{L}}_{\mathfrak{D}}(\theta) - \hat{\mathcal{L}}_{\bar{\mathfrak{D}}}(\theta) \right| \leq \sum_{h'=1}^{H} \sum_{j'=1}^{T} c_{(h',j')} \mathbb{I}\left[ \xi_{h'}^{(j')} = \bar{\xi}_{h'}^{(j')} \right] \tag{6}$$

Intuitively, $c_{(h,j)}$ quantifies the *transition-level influence* of altering the $(h, j)$-th state–action–reward tuple on the average return. A complete derivation—including a justification of why this bound covers propagation of the perturbed transition to future steps—is given in Appendix B.2. The result yields the explicit vector $c$ used in the main Theorem 3.2.

$$c_{(h,t)} = \frac{\gamma^{h-1} R_{\max}}{T}, \qquad \|c\|^2 = \frac{R_{\max}^2}{T(1-\gamma^2)}\left(1 - \gamma^{2H}\right). \tag{7}$$

Combining these with McDiarmid's inequality for Markov chains gives a **trajectory-dependent tail bound** on the deviation $\mathcal{L}(\theta) - \hat{\mathcal{L}}_{\mathfrak{D}}(\theta)$. With standard PAC-Bayes bound derivation we obtain our primary result, a PAC-Bayesian value-error bound for MDPs in Theorem 3.2:

**Theorem 3.2** *Let the reward function be bounded in $[0, R_{max}]$ and let $\mathcal{M}$ be a (not necessarily time-homogeneous) Markov Decision Process (MDP) induced by any policy $\pi_\theta$ such that it satisfies $\tau_{\min} < \infty$. For any prior $\mu$ over $\Pi$, any posterior $\rho$ chosen after interacting with the environment, and any $\delta \in (0, 1)$, with probability at least $1 - \delta$ over the sample $\mathfrak{D}$ of $T$ trajectories with time horizon $H$:*

$$\mathbb{E}_{\theta \sim \rho} \left[ \mathcal{L}(\theta) - \hat{\mathcal{L}}_{\mathfrak{D}}(\theta) \right] \leq \sqrt{\frac{R_{\max}^2 \, \tau_{\min} \left( 1 - \gamma^{2H} \right)}{2T(1 - \gamma^2)} \left( \mathrm{KL}(\rho \| \mu) + \ln \frac{2}{\delta} \right)}. \tag{8}$$

where $\tau_{\min}$ is the *mixing time* of the chain, the smallest number of steps after which the distribution of the chain's state is, in a statistical sense, nearly indistinguishable from its long-run or stationary distribution in Total Variation distance, no matter where the chain started. In other words, it measures how quickly the chain "forgets" its initial state and becomes well mixed.

The bound in 3.2 can be straightforwardly converted to a PAC-Bayes bound on the error of a value function $V_{\pi_\theta}$ for a policy $\pi_\theta$, using the fact that $\mathcal{L}(\theta) = -V_{\pi_\theta}$ (5):

$$\mathbb{E}_{\theta \sim \rho} [V_{\pi_\theta}] \geq - \mathbb{E}_{\theta \sim \rho} \left[ \hat{\mathcal{L}}_{\mathfrak{D}}(\theta) \right] - \sqrt{\frac{R_{\max}^2 \, \tau_{\min} \left( 1 - \gamma^{2H} \right)}{2T(1 - \gamma^2)} \left( \mathrm{KL}(\rho \| \mu) + \ln \frac{2}{\delta} \right)}. \tag{9}$$

One may notice that it has a structure remarkably similar to Upper Confidence Bounds (UCB) [32] used in bandit algorithms $Q(a) \leq \hat{Q}(a) + U_t(a)$, where the true value is bounded by an empirical estimate plus an uncertainty term. In our case, the uncertainty term accounts for three key factors: **(1)** the statistical challenge of working with limited trajectory data, addressed by the $\frac{1}{T}$ term; **(2)** the temporal correlation structure of the MDP, captured by $\tau_{\min}$ and the discount-related terms; and **(3)** the complexity of the policy class, represented by the KL divergence.

This UCB-like interpretation suggests a natural approach to policy optimization: select the posterior $\rho$ that maximizes this lower bound. Such a strategy would automatically balance exploitation (maximizing the empirical value) and theoretically-justified exploration (accounting for uncertainty). This is precisely the approach implemented in the PB-SAC algorithm, where we periodically optimize the posterior distribution and inject its knowledge back into the policy to guide learning.

**From theory to practice.** Theorem 3.2 provides a high-confidence guarantee on the difference between empirical and true returns for stochastic policies, using three key components: the posterior–prior KL divergence, the squared coefficient vector $\|c\|^2$ from (7), and the mixing time $\tau_{\min}$. The central question becomes how to leverage this certificate to enhance learning. In Section 3.1, we demonstrate that this bound can be actively optimized during training by integrating it within a modern actor–critic framework. The resulting procedure, *PB-SAC*, transforms our theoretical guarantee into a principled approach for balancing exploration and exploitation while maintaining formal certificates on policy performance.

## 3.1 A Practical Algorithm based on PAC-Bayes RL

Our algorithm, **PAC-Bayes-Certified Soft Actor–Critic (*PB-SAC*)**, operationalizes the PAC-Bayes value-error bound of Theorem 3.2 within a Soft Actor-Critic (SAC) training loop. Building upon the periodic update cycle described earlier, PB-SAC maintains a posterior distribution over policy parameters and injects sampled knowledge to guide exploration. While it shares the "distribution-over-policies" principle with EPICG and EPICG-SAC [29], our approach differs in three fundamental ways: (i) It focuses on **single-task optimization** rather than task streams, though the framework naturally extends to Lifelong RL scenarios. (ii) It explicitly optimizes **the exact value-error bound** from Theorem 3.2 during training and **monitors the bound value** at each update cycle, providing

continuous performance guarantees. (iii) It applies the posterior distribution **directly to actor parameters** $\theta$ rather than relying on critic ensembles or Bellman-error surrogates as in PBAC, resulting in a more streamlined and computationally efficient implementation. Additionally, PBAC [14] trains critics via a Catoni-type Bellman-error bound but never reports the bound value; it is used only as a loss. EPICG/EPICG-SAC of [29] regularizes the KL between posterior and running prior yet does not compute or log the PAC-Bayes bound either.

**Posterior, prior, and sampling.** Let $\theta$ denote the flattened parameters of the actor network. The *posterior* $\rho$ is a diagonal Gaussian, $\theta \sim \mathcal{N}(\upsilon, \mathrm{diag}(\sigma^2))$, where both the mean $\upsilon$ and standard deviation $\sigma$ are learnable, gradient-tracked variables. The *prior* $\mu$ is simply a copy of the posterior at initialization, or after a PAC-Bayes update cycle; this choice maintains meaningful guarantees by preventing $\mathrm{KL}(\rho\|\mu)$ from exploding while ensuring that the core SAC algorithm can continue learning. Additionally, without this resetting mechanism, the $\mathrm{KL}(\rho\|\mu)$ regularization would permanently penalize deviations from the initial actor parameters, significantly hindering the learning process.

**PAC-Bayes update cycle.** Our algorithm performs periodic PAC-Bayes updates using completely new data batches to maintain theoretical guarantees. At each update point, we freeze the current policy and collect a fresh batch of $T$ trajectories, which are used exclusively for the current PAC-Bayes analysis. Using this batch, we fit a posterior distribution $\rho$ by optimizing $\mathcal{L}_\rho = \mathbb{E}_{\theta\sim\rho}[-Q(s, \pi_\theta(s))] + \beta\mathrm{KL}(\rho\|\mu)$, balancing critic values against divergence from the prior. We estimate the mixing time $\tau_{\min}$ from trajectory autocorrelations and compute the PAC-Bayes bound according to Theorem 3.2. Crucially, we then set the prior for the next update cycle to the current posterior, $\mu_{\mathrm{new}} := \rho$, creating a "checkpoint" that preserves the bound's validity while allowing continued learning. Before resuming training, we inject knowledge from the posterior by sampling parameters $\theta_{\mathrm{sampled}} \sim \rho$ and mixing them with the current policy: $\theta_{\mathrm{new}} = \lambda\theta_{\mathrm{sampled}} + (1 - \lambda)\theta_{\mathrm{current}}$. Each update's data is then discarded, ensuring that no data point influences multiple bounds, thereby maintaining the theoretical guarantees of our approach. The pseudo-code 1 bellow shows the full training loop which interleaves standard SAC updates with these PAC-Bayes updates.

**Why the bound is practically non-vacuous.** For typical values of $\gamma$ ($\simeq 0.99$), the classical bound of Fard *et al.* [13] suffers from the constraint on the number of samples needed $H > R^4/(1 - \gamma)^4$, while The bound of Tasdighi *et al.* [14] requires the same amount to beat triviality—a number that themselves flag as "rarely fulfilled in practice". Our transition-level analysis shrinks the bound to $T > \frac{R_{\max}^2 \, \tau_{\min}\left(1-\gamma^{2H}\right)}{2(1-\gamma^2)}$ (for long trajectories $(1 - \gamma^{2H}) \simeq 1$). Although this might appear costly due to the dependence on the number of trajectories, it is in fact substantially more tractable than the classical bound (the power of 2 is only on $\gamma$ not $1 - \gamma$). In practice, the bound can be reduced even further: by obtaining a rough estimate of the mixing time $\tau_{\min}$, one can choose $H$ to be just above this threshold. This keeps the term $(1 - \gamma^{2H})$ in the numerator below one, tightening the bound. As a result, rather than requiring long trajectories, it suffices to collect many short ones. Further discussion can be found in Appendix B.6

# 4 Experimental Setup

To evaluate our PAC-Bayesian reinforcement learning approach, we utilize the ICU-Sepsis environment [33], a benchmark MDP built from real medical data that simulates sepsis treatment in intensive care units. This environment represents an important real-world sequential decision-making problem with significant healthcare implications. ICU-Sepsis is a tabular MDP with 716 discrete states representing different patient conditions and 25 possible actions corresponding to various combinations of medical interventions, primarily focusing on intravenous fluid and vasopressor dosages. Each episode simulates a patient's treatment journey, where the agent (representing the clinician) observes the patient's state, selects appropriate interventions, and then observes how the patient's condition evolves in response to treatment. The environment uses a reward structure where survival results in a terminal reward of $+1$, while death corresponds to a reward of $0$. All intermediate rewards are also 0, making the expected return equivalent to the probability of patient survival.

**Algorithms and Implementation.** We compare our PAC-Bayes-certified Soft Actor-Critic (PB-SAC)[1] against standard baselines. PB-SAC extends SAC with the PAC-Bayesian framework from Section 3, maintaining a posterior over policy parameters to provide high-probability performance guarantees. Our baselines include SAC (off-policy maximum entropy), DQN (value-based), and PPO (on-policy). All implementations use tabular representations for ICU-Sepsis, appropriate for its discrete state-action space. Additionally We also evaluate on continuous control MuJoCo benchmarks [34, 35] (Walker2d-v5, Humanoid-v5, HalfCheetah-v5) to assess scalability to continuous domains, sample efficiency, and generalization of performance guarantees. *Our protocol* consists of running each algorithm for 300,000 episodes across 5-10 random seeds. For PB-SAC, we perform periodic PAC-Bayes updates every 20,000 steps to maintain the posterior distribution and compute certified bounds. We evaluate the algorithms using two primary metrics. The first is certified performance, defined as a high-probability lower bound on return, holding with probability $1 - \delta$. The second is average return, which corresponds to the expected survival probability. This evaluation framework allows for a rigorous and comprehensive comparison across both clinical and standard continuous-control benchmarks.

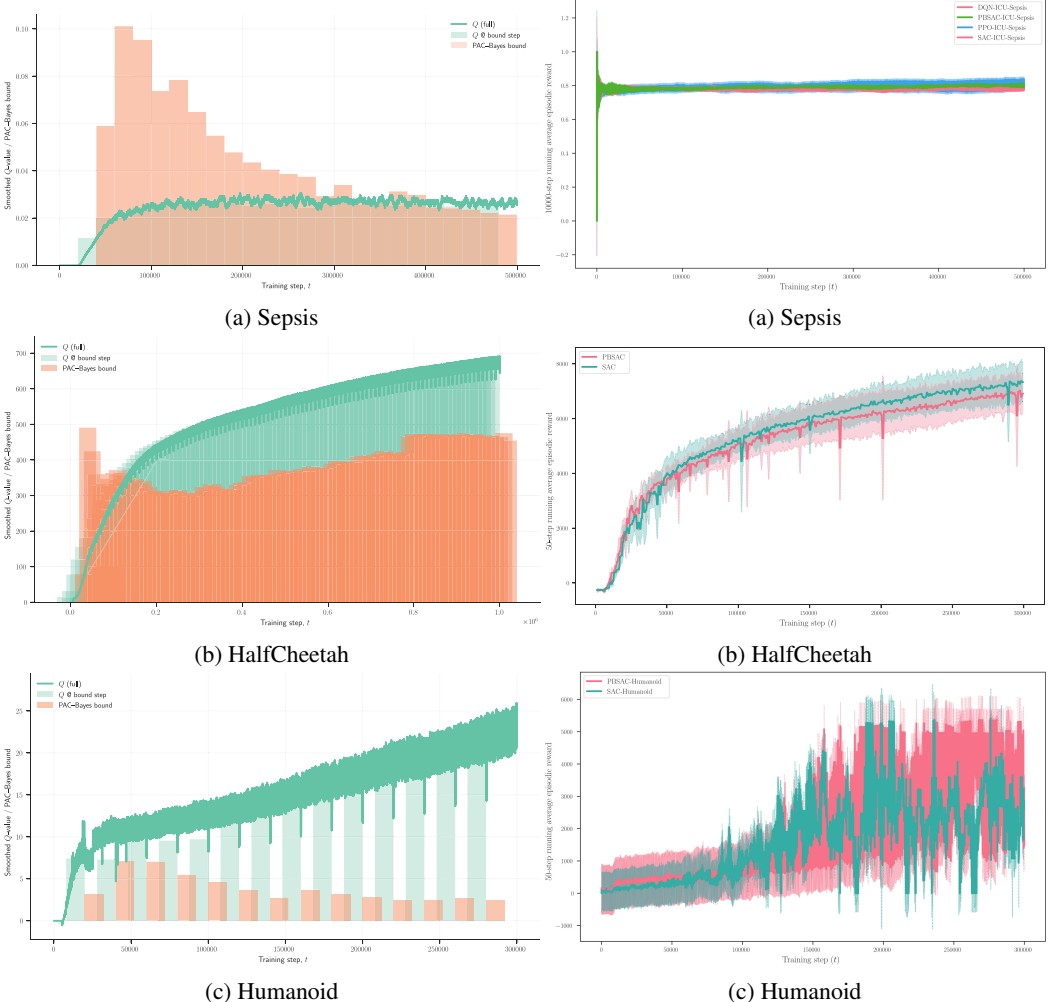

(a) Sepsis

(a) Sepsis

(b) HalfCheetah

(b) HalfCheetah

(c) Humanoid

(c) Humanoid

Figure 1: PAC-Bayes bounds (green bars) *vs.* Q-values (orange bars), along with the running average of the empirical Q-values (green line). A tight and desirable lower bound is one that closely approaches the Q-function.

Figure 2: Running average of episodic returns: A comparison between our PB-SAC (shown in pink in (b) and (c), and in green in (a)) and its base algorithm (SAC alone).

---

[1]The code can be found here `https://anonymous.4open.science/r/BenchRL-72B7`

## 4.1 Empirical Analysis

Figures 1 and 2 demonstrate **PB-SAC**'s performance compared to standard SAC across three environments. Figure 1 (left column) illustrates the evolution of PAC-Bayes certificates (green) relative to learned $Q$-values (orange), while Figure 2 traces episodic returns throughout training. Three distinct patterns emerge from these results. First, **certificates tighten at environment-dependent rates**; In the Sepsis environment, guarantees become informative almost immediately after training begins and remain closely coupled to the critic thereafter. This rapid tightening stems primarily from the simulator's quick mixing dynamics, which benefit our PAC-Bayes bound, additional data rapidly sharpens the certificate. For **HalfCheetah**, the bound improves more gradually, reflecting the environment's longer effective horizon where states remain correlated over extended periods. Finally, **Humanoid** represents the most challenging scenario with its numerous degrees of freedom and complex dynamics. Although guarantees become comparatively looser in this environment, they consistently remain informative, never deteriorating to the trivial bound of zero.

**Guided exploration without sacrificing reward.** Turning to Fig. 2, PB-SAC matches the baselines on Sepsis, keeps pace on HalfCheetah, and modestly outperforms SAC on Humanoid. As the algorithm injects posterior samples selected by the bound, it explores in directions that carry provable upside—yet the additional regularization never derails learning. The results suggest that safety certificates and competitive return need not be at odds. Across tasks we observe a clear narrative: the faster the environment mixes, the faster the PAC-Bayes certificate closes the gap to the critic. This empirical pattern echoes the explicit mixing-time factor in Theorem 3.2 and underscores why reporting an estimate of $\tau_{\min}$ can contextualize confidence results. We therefore recommend including mixing-time diagnostics in future evaluations of safe RL methods.

In summary, PB-SAC converts a theoretically principled bound into a live learning signal: it produces meaningful confidence certificates early, preserves or improves return, and exhibits behavior that aligns with the qualitative dependence on Markov mixing predicted by our analysis.

## 5   Conclusion and Limitations

In this work, we introduced PB-SAC, a PAC-Bayesian actor–critic algorithm that advances the intersection of reinforcement learning and Bayesian guarantees. If adapted to the EPICG/EPICG-SAC framework [29], we believe PB-SAC holds the potential to address the well-known *plasticity–stability* dilemma, a prominent research challenge in lifelong and continual reinforcement learning. This extension would allow the agent to balance the retention of useful prior knowledge (*stability*) with the acquisition of new information (*plasticity*), using PAC-Bayesian guarantees as a principled mechanism for managing uncertainty. Empirically, PB-SAC matches or surpasses SAC on both clinical and continuous-control benchmarks, while yielding confidence bounds that tighten predictably with the environment's mixing time. This marks a step forward toward certified reinforcement learning algorithms suitable for real-world deployment, particularly in high-stakes domains where reliable performance guarantees are essential.

**Limitations.** Our current framework uses a Kullback–Leibler (KL) divergence penalty between prior and posterior over policy parameters. Although KL is analytically convenient, it does not respect the intrinsic geometry of the parameter space and can exhibit unstable behavior when distributions diverge significantly [36]. In high-dimensional settings, computing KL gradients is computationally intensive and may force the posterior to collapse onto the prior—resulting in overly constrained updates that hinder meaningful learning progress [37].

An appealing alternative is to employ a Wasserstein distance within the PAC-Bayes bound. Recent work has developed high-probability PAC-Bayesian inequalities based on Wasserstein metrics, which naturally capture distributional geometry and avoid degenerate update regimes even when supports are disjoint [38, 39, 36]. Moreover, entropic (Sinkhorn) smoothing enables scalable stochastic variational inference under Wasserstein regularization, making posterior updates tractable in high dimensions [40]. Incorporating a Wasserstein-based PAC-Bayes bound and a corresponding Sinkhorn-SVI scheme is a key direction for future research, with the potential to yield tighter certificates and more robust policy learning.

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

# A  Mathematical Tools

**Lemma A.1 (Markov's Inequality)** *For any random variable $X$ such that $\mathbb{E}[|X|] = \mu$, for any $a > 0$, we have*

$$\mathbb{P}\{|X| \geq a\} \leq \frac{\mu}{a}.$$

**Lemma A.2 (Change of measure)** *For any measurable function $f : \Theta \to \mathbb{R}$ and distributions $\mu, \rho \in \mathcal{P}(\Theta)$:*

$$\mathbb{E}_{\theta \sim \rho}[f(\theta)] \leq \mathrm{KL}(\rho \| \mu) + \ln \mathbb{E}_{\theta \sim \mu}[\exp(f(\theta))] \tag{10}$$

*where $\mathrm{KL}(\rho \| \mu)$ is the Kullback-Leibler divergence between $\rho$ and $\mu$.*

## A.1  Concentration for Markov chains via Marton coupling

We use Paulin [15]'s extension of McDiarmid's bounded-difference inequality to Markov chains. This extension provides concentration inequalities for functions of dependent random variables, with constants that depend on the mixing properties of the chain.

### A.1.1  Marton coupling and mixing time

The key insight in Paulin's [15] approach is to use a coupling structure known as Marton coupling, which quantifies the dependency between random variables in a Markov chain. For a Markov chain $X = (X_1, \ldots, X_N)$ on state space $\Lambda = \Lambda_1 \times \ldots \times \Lambda_N$, a Marton coupling provides a way to couple the distributions of future states conditioned on different past states.

Let $\tau(\varepsilon)$ denote the mixing time of the chain $X$ in total variation distance, defined as the minimal $t$ such that for every $1 \leq i \leq N - t$ and $x, y \in \Lambda_i$:

$$d_{TV}(\mathcal{L}(X_{i+t}|X_i = x), \mathcal{L}(X_{i+t}|X_i = y)) \leq \varepsilon \tag{11}$$

We define the normalized mixing time parameter $\tau_{\min}$ as:

$$\tau_{\min} = \inf_{0 \leq \varepsilon < 1} \tau(\varepsilon) \left( \frac{2-\varepsilon}{1-\varepsilon} \right)^2 \tag{12}$$

 **A.1.2 McDiarmid's inequality for Markov chains**

 For a function $f(X)$ satisfying the bounded-differences property: for any $x, y \in \Lambda$,

$$f(x) - f(y) \leq \sum_{i=1}^{N} c_i \mathbb{I}[x_i \neq y_i] \tag{13}$$

 where $c \in \mathbb{R}_+^N$ and $\mathbb{I}[\text{condition}]$ is the indicator function, Paulin's theorem gives:

$$\Pr\big(|f(X) - \mathbb{E}f(X)| \geq t\big) \leq 2\exp\big(-2t^2/\|c\|^2 \tau_{\min}\big). \tag{14}$$

 The norm $\|c\|^2$ is defined as $\sum_{i=1}^{N} c_i^2$.

 **A.1.3 Application to bounded differences in MDPs**

 For Markov decision processes, this inequality is particularly useful when analyzing the difference
 between value functions. If perturbing a single transition can change the value by at most $c_i$, then the
 total effect on a function of trajectories is bounded by the above concentration inequality, with the
 mixing time of the MDP properly accounting for the propagation of the perturbation through future
 states.

 # B Derivation of PAC-Bayes Value-Error Bound for RL

 ## B.1 Bounded-differences property for MDP trajectories

 We begin by recalling the definitions of discounted return for a trajectory $\xi$ and the corresponding
 value function from Section 3:

$$G(\xi) = \sum_{k=0}^{H-1} \gamma^k R_{k+1}$$
$$V_{\pi_\theta} = \mathbb{E}_{\xi \sim \mathcal{M}}[G(\xi)]$$

 As defined in equation (5), our empirical and expected losses are:

$$\hat{\mathcal{L}}_{\mathfrak{D}}(\theta) = -\frac{1}{T}\sum_{j=1}^{T} G(\xi^{(j)})$$
$$\mathcal{L}(\theta) = -\mathbb{E}_{\xi \sim \mathcal{M}}[G(\xi)] = -V_{\pi_\theta}$$

 To apply McDiarmid's inequality for Markov chains, we must establish the bounded-differences
 condition for our empirical loss. Specifically, we need to show that replacing one transition in a
 trajectory affects $\hat{\mathcal{L}}_{\mathfrak{D}}(\theta)$ by at most $\sum_{h=1}^{H}\sum_{j=1}^{T} c_{(h,j)}\mathbb{I}[\xi_h^{(j)} \neq \bar{\xi}_h^{(j)}]$, where $c \in \mathbb{R}_+^{H \times T}$ and $\mathbb{I}$ is the
 indicator function.

 ## B.2 Quantifying the impact of perturbed transitions

 Suppose we replace a single transition at position $h$ in trajectory $j$. The change in the discounted
 return of this trajectory is bounded by:

$$|G(\xi^{(j)}) - G(\bar{\xi}^{(j)})| = |\gamma^{h-1}(R_h - \bar{R}_h) + \text{effects on future rewards}|$$
$$\leq \gamma^{h-1} R_{\max} + \text{effects on future rewards}$$

 Crucially, this perturbation affects not only the immediate reward but potentially all subsequent
 transitions and rewards in that trajectory. The change in our empirical loss is therefore bounded by:

$$|\hat{\mathcal{L}}_{\mathfrak{D}}(\theta) - \hat{\mathcal{L}}_{\bar{\mathfrak{D}}}(\theta)| \leq \frac{\gamma^{h-1} R_{\max}}{T} = c_{(h,j)}$$

### B.3 Derivation of $\|c\|^2$ for the PAC-Bayes bound

To apply McDiarmid's inequality for Markov chains as developed by [15], we need to compute $\|c\|^2$:

$$\|c\|^2 = \sum_{j=1}^{T} \sum_{h=1}^{H} c^2(h,j)$$

$$= \frac{R_{\max}^2}{T^2} \cdot T \sum_{h=1}^{H} \gamma^{2(h-1)}$$

$$= \frac{R_{\max}^2}{T} \underbrace{\sum_{h=0}^{H-1} \gamma^{2h}}_{\text{finite geometric series}}$$

$$= \frac{R_{\max}^2}{T} \cdot \frac{1 - \gamma^{2H}}{1 - \gamma^2}.$$

For infinite-horizon settings where $H \to \infty$ and $\gamma < 1$, the series converges to $1/(1 - \gamma^2)$, this simplifies to

$$\|c\|^2 = \frac{R_{\max}^2}{T(1 - \gamma^2)}.$$

### B.4 Full accounting of perturbation propagation effects

A critical question is whether our derivation of $\|c\|^2$ fully accounts for the propagation of perturbations through the trajectory. Since a perturbation at step $h$ in trajectory $j$ affects all subsequent transitions in that trajectory, the bounded-differences indicator is 1 for every $(h', j)$ with $h' \geq h$.

For a perturbation at step $h$ in trajectory $j$, the sum of corresponding coefficients is:

$$\sum_{h'=h}^{H} c_{(h',j)} = \frac{R_{\max}}{T} \sum_{k=0}^{H-h} \gamma^{h-1+k} = \frac{R_{\max} \gamma^{h-1}}{T} \cdot \frac{1 - \gamma^{H-h+1}}{1 - \gamma}$$

The actual maximum change in discounted return from this perturbation (worst case: reward changes from 0 to $R_{\max}$) is:

$$|G(\xi^{(j)}) - G(\bar{\xi}^{(j)})| \leq R_{\max} \gamma^{h-1} \sum_{k=0}^{H-h} \gamma^k = R_{\max} \gamma^{h-1} \cdot \frac{1 - \gamma^{H-h+1}}{1 - \gamma}$$

When divided by $T$ (because $\hat{\mathcal{L}}_{\mathfrak{D}}(\theta)$ averages over $T$ trajectories), we get exactly the same quantity as the sum of coefficients above. Therefore, the bounded-differences condition holds with equality, confirming that our derivation of $\|c\|^2$ fully accounts for all propagation effects without requiring additional constants.

This careful accounting of propagation effects allows us to apply McDiarmid's inequality for Markov chains to obtain the PAC-Bayes bound in Theorem 3.2 with the correct constants.

### B.5 Derivation of the PAC-Bayes Bound

Having established the bounded-differences property and quantified the impact of perturbations via $\|c\|^2$, we now derive the PAC-Bayes bound on the expected difference between empirical and true losses.

### B.5.1 From McDiarmid to moment generating function

McDiarmid's inequality for Markov chains (Equation (14)) provides a concentration inequality on the deviation between empirical and expected losses. From this, we can derive a bound on the moment generating function (MGF) as shown by [15]:

**Lemma B.1 (MGF bound for Markov chains)** *For any $\lambda > 0$ and policy parameters $\theta \in \Theta$:*

$$\mathbb{E}_{\mathfrak{D} \sim \mathcal{M}^{(T)}} \left[ \exp \left( \lambda(\hat{\mathcal{L}}_{\mathfrak{D}}(\theta) - \mathcal{L}(\theta)) \right) \right] \leq \exp \left( \frac{\lambda^2 \|c\|^2 \tau_{\min}}{8} \right) \tag{15}$$

*where $\tau_{\min}$ is the mixing time of the Markov chain induced by policy $\pi_\theta$.*

### B.5.2 PAC-Bayes change of measure

Now we can follow the standard PAC-Bayes derivation. Let $\Theta$ be our parameter space and let $\mu \in \mathcal{P}(\Theta)$ be a prior distribution over $\Theta$ chosen independently of the data. For any posterior distribution $\rho \in \mathcal{P}(\Theta)$ (which may depend on $\mathfrak{D}$), we apply the change-of-measure inequality (Donsker–Varadhan [41] variational formula)

Let $f(\theta) = \lambda(\hat{\mathcal{L}}_{\mathfrak{D}}(\theta) - \mathcal{L}(\theta))$. Applying Lemma A.2:

$$\mathbb{E}_{\theta \sim \rho}[\lambda(\hat{\mathcal{L}}_{\mathfrak{D}}(\theta) - \mathcal{L}(\theta))] \leq \mathrm{KL}(\rho \| \mu) + \ln \mathbb{E}_{\theta \sim \mu}[\exp(\lambda(\hat{\mathcal{L}}_{\mathfrak{D}}(\theta) - \mathcal{L}(\theta)))] \tag{16}$$

### B.5.3 Combining with the MGF bound

Taking the expectation with respect to $\mathfrak{D} \sim \mathcal{M}^{(T)}$ on both sides:

$$\mathbb{E}_{\mathfrak{D}} \mathbb{E}_{\theta \sim \rho}[\lambda(\hat{\mathcal{L}}_{\mathfrak{D}}(\theta) - \mathcal{L}(\theta))] \leq \mathrm{KL}(\rho \| \mu) + \mathbb{E}_{\mathfrak{D}} \ln \mathbb{E}_{\theta \sim \mu}[\exp(\lambda(\hat{\mathcal{L}}_{\mathfrak{D}}(\theta) - \mathcal{L}(\theta)))] \tag{17}$$

By Jensen's inequality, since $\ln$ is concave:

$$\mathbb{E}_{\mathfrak{D}} \mathbb{E}_{\theta \sim \rho}[\lambda(\hat{\mathcal{L}}_{\mathfrak{D}}(\theta) - \mathcal{L}(\theta))] \leq \mathrm{KL}(\rho \| \mu) + \ln \mathbb{E}_{\mathfrak{D}} \mathbb{E}_{\theta \sim \mu}[\exp(\lambda(\hat{\mathcal{L}}_{\mathfrak{D}}(\theta) - \mathcal{L}(\theta)))] \tag{18}$$

By Fubini's theorem (exchanging the order of expectations) and Lemma B.1:

$$\mathbb{E}_{\mathfrak{D}} \mathbb{E}_{\theta \sim \rho}[\lambda(\hat{\mathcal{L}}_{\mathfrak{D}}(\theta) - \mathcal{L}(\theta))] \leq \mathrm{KL}(\rho \| \mu) + \ln \mathbb{E}_{\theta \sim \mu} \mathbb{E}_{\mathfrak{D}}[\exp(\lambda(\hat{\mathcal{L}}_{\mathfrak{D}}(\theta) - \mathcal{L}(\theta)))] \tag{19}$$

$$\leq \mathrm{KL}(\rho \| \mu) + \ln \mathbb{E}_{\theta \sim \mu} \left[ \exp \left( \frac{\lambda^2 \|c\|^2 \tau_{\min}}{8} \right) \right] \tag{20}$$

$$= \mathrm{KL}(\rho \| \mu) + \frac{\lambda^2 \|c\|^2 \tau_{\min}}{8} \tag{21}$$

Dividing by $\lambda > 0$:

$$\mathbb{E}_{\mathfrak{D}} \mathbb{E}_{\theta \sim \rho}[\hat{\mathcal{L}}_{\mathfrak{D}}(\theta) - \mathcal{L}(\theta)] \leq \frac{\mathrm{KL}(\rho \| \mu)}{\lambda} + \frac{\lambda \|c\|^2 \tau_{\min}}{8} \tag{22}$$

### B.5.4 High-probability bound via Markov's inequality

Now, we convert this expectation bound into a high-probability bound. By Markov's inequality A.1, for any non-negative random variable $X$ and $\delta > 0$:

With probability at least $1 - \delta$:

$$\mathbb{E}_{\theta \sim \rho}[\hat{\mathcal{L}}_{\mathfrak{D}}(\theta) - \mathcal{L}(\theta)] \leq \frac{\mathrm{KL}(\rho \| \mu) + \ln \frac{2}{\delta}}{\lambda} + \frac{\lambda \|c\|^2 \tau_{\min}}{8} \tag{23}$$

### B.5.5 Optimizing the bound

To tighten the bound, we minimize the right-hand side with respect to $\lambda > 0$. Taking the derivative and setting it to zero:

$$\frac{\partial}{\partial \lambda} \left( \frac{\mathrm{KL}(\rho\|\mu) + \ln \frac{2}{\delta}}{\lambda} + \frac{\lambda \|c\|^2 \tau_{\min}}{8} \right) = 0 \tag{24}$$

$$-\frac{\mathrm{KL}(\rho\|\mu) + \ln \frac{2}{\delta}}{\lambda^2} + \frac{\|c\|^2 \tau_{\min}}{8} = 0 \tag{25}$$

Solving for the optimal $\lambda^*$:

$$\lambda^* = \sqrt{\frac{8(\mathrm{KL}(\rho\|\mu) + \ln \frac{2}{\delta})}{\|c\|^2 \tau_{\min}}} \tag{26}$$

Substituting $\lambda^*$ back into our bound:

$$\mathbb{E}_{\theta \sim \rho}[\hat{\mathcal{L}}_{\mathfrak{D}}(\theta) - \mathcal{L}(\theta)] \leq \frac{\mathrm{KL}(\rho\|\mu) + \ln \frac{2}{\delta}}{\lambda^*} + \frac{\lambda^* \|c\|^2 \tau_{\min}}{8} \tag{27}$$

$$= \sqrt{\frac{\|c\|^2 \tau_{\min}(\mathrm{KL}(\rho\|\mu) + \ln \frac{2}{\delta})}{8}} + \sqrt{\frac{\|c\|^2 \tau_{\min}(\mathrm{KL}(\rho\|\mu) + \ln \frac{2}{\delta})}{8}} \tag{28}$$

$$= \sqrt{\frac{\|c\|^2 \tau_{\min}(\mathrm{KL}(\rho\|\mu) + \ln \frac{2}{\delta})}{2}} \tag{29}$$

### B.5.6 Final bound

Finally, substituting the expression for $\|c\|^2$ from Equation (7):

$$\mathbb{E}_{\theta \sim \rho}[\hat{\mathcal{L}}_{\mathfrak{D}}(\theta) - \mathcal{L}(\theta)] \leq \sqrt{\frac{\frac{R_{\max}^2}{T} \cdot \frac{1 - \gamma^{2H}}{1 - \gamma^2} \cdot \tau_{\min} \cdot (\mathrm{KL}(\rho\|\mu) + \ln \frac{2}{\delta})}{2}} \tag{30}$$

$$= \sqrt{\frac{R_{\max}^2 \tau_{\min}(1 - \gamma^{2H})}{2T(1 - \gamma^2)} \left( \mathrm{KL}(\rho\|\mu) + \ln \frac{2}{\delta} \right)} \tag{31}$$

Recalling that $\mathcal{L}(\theta) = -V_{\pi_\theta}$ from Equation (5), we obtain the PAC-Bayes value-error bound stated in Theorem 3.2:

$$\mathbb{E}_{\theta \sim \rho}[V_{\pi_\theta}] \geq \mathbb{E}_{\theta \sim \rho}[-\hat{\mathcal{L}}_{\mathfrak{D}}(\theta)] - \sqrt{\frac{R_{\max}^2 \tau_{\min}(1 - \gamma^{2H})}{2T(1 - \gamma^2)} \left( \mathrm{KL}(\rho\|\mu) + \ln \frac{2}{\delta} \right)} \tag{32}$$

This bound provides a high-probability lower bound on the expected value of policies sampled from the posterior distribution $\rho$, accounting for the statistical dependencies inherent in MDP trajectories through the mixing time $\tau_{\min}$.

## B.6 A discussion on Tasdighi et al.'s assumption

An assumption that is worth noting in the work of Tasdighi et al. [14] is that the sequence of Bellman errors forms a Markov chain. Here, we provide a simple counter-example that demonstrates why this assumption does not hold in general.

Consider a simple MDP with four states $\{A, B, C, D\}$ and the following transition dynamics with discount factor $\gamma = 0$:

- State $A$ transitions to state $C$ with reward $r = 0$
- State $B$ transitions to state $D$ with reward $r = 0$
- State $C$ has a self-loop with reward $r = +1$
- State $D$ has a self-loop with reward $r = -1$

Let us use a value function $V \equiv 0$ that assigns zero value to all states. We can then compute the Bellman errors for each state:

$$\delta_t(A) = r(A) + \gamma \max_a \mathbb{E}[V(s')|s = A, a] - V(A) = 0 + 0 \cdot V(C) - 0 = 0 \qquad (33)$$

$$\delta_t(B) = r(B) + \gamma \max_a \mathbb{E}[V(s')|s = B, a] - V(B) = 0 + 0 \cdot V(D) - 0 = 0 \qquad (34)$$

Thus, both states $A$ and $B$ produce the same Bellman error $\delta_t = 0$ at time $t$. However, the subsequent Bellman errors at time $t + 1$ are:

$$\delta_{t+1}(C) = r(C) + \gamma \max_a \mathbb{E}[V(s')|s = C, a] - V(C) = +1 + 0 \cdot V(C) - 0 = +1 \qquad (35)$$

$$\delta_{t+1}(D) = r(D) + \gamma \max_a \mathbb{E}[V(s')|s = D, a] - V(D) = -1 + 0 \cdot V(D) - 0 = -1 \qquad (36)$$

This simple example demonstrates that knowing the current Bellman error $\delta_t = 0$ is insufficient to determine the distribution of the next Bellman error $\delta_{t+1}$, which can be either $+1$ or $-1$ depending on the state that produced the current error.

For a sequence to be Markovian, the conditional distribution of future states must depend only on the current state, not on the sequence of events that preceded it. In this case, the distribution of $\delta_{t+1}$ depends on which state ($A$ or $B$) produced $\delta_t = 0$, not just on the value of $\delta_t$ itself.

Therefore, the sequence of Bellman errors $\{\delta_t\}$ cannot be modeled as a Markov chain in general, invalidating a key assumption in the theoretical analysis of Tasdighi et al. [14].

## C  Pseudo Code

---

**Algorithm 1:** PAC-Bayes Soft Actor-Critic *(PB-SAC)*

---

**Input:** MDP $\mathcal{M} = (\mathcal{S}, \mathcal{A}, \mathbb{P}, R, \gamma)$, discount $\gamma$, failure probability $\delta$, KL coefficient $\beta$, mixing coefficient $\lambda$

**Output:** Policy $\pi_\theta$ with PAC-Bayes guarantees

```
/* Initialize                                                    */
```
1  Initialize SAC components (actor $\pi_\theta$, critics $Q_{\phi_1}, Q_{\phi_2}$, replay buffer $\mathcal{D}$)
2  Initialize posterior $\rho(\theta) = \mathcal{N}(\upsilon, \mathrm{diag}(\sigma^2))$ where $\upsilon$ are the initial actor parameters
3  Initialize prior $\mu(\theta) = \mathcal{N}(\upsilon, \mathrm{diag}(\sigma^2))$
4  Initialize PAC-Bayes rollout sizes (horizon $H$, trajectories $T$)
5  $r_{\max} \leftarrow$ initial_estimate; $\tau_{\min} \leftarrow 1$

6  **for** $t = 1$ **to** *total_timesteps* **do**
7  |  $a_t \leftarrow \pi_\theta(s_t)$                                      `// Standard action selection`
8  |  $s_{t+1}, r_t, \mathrm{done}_t \leftarrow$ env.step$(a_t)$
9  |  Store $(s_t, a_t, r_t, s_{t+1}, \mathrm{done}_t)$ in $\mathcal{D}$
10 |  $r_{\max} \leftarrow \max(r_{\max}, r_t)$
   |  ```/* Standard SAC update                                      */```
11 |  Update actor and critic networks using SAC
12 |  $\theta_{\mathrm{current}} \leftarrow$ _flatten_policy_params$(\pi_\theta)$ ```/* PAC-Bayes updates (infrequent)    */```
13 |  **if** $t \bmod$ *pb_update_freq* $= 0$ **then**
14 |  |  $\mathfrak{D} \leftarrow$ collect_rollouts$(T, H)$                    `// Collect trajectories`
   |  |  ```/* Update posterior                                      */```
15 |  |  Sample states $\{s^{(i)}\}$ from $\mathfrak{D}$
16 |  |  Optimize $\mathbb{E}_{\theta \sim \rho}\left[\hat{\mathcal{L}}_{\mathfrak{D}}(\theta)\right] + \beta\, \mathrm{KL}$
   |  |  ```/* Compute PAC-Bayes bound                               */```
17 |  |  $\tau_{\min} \leftarrow$ estimate_mixing_time$(\mathfrak{D})$
18 |  |  bound $\leftarrow \sqrt{\frac{r_{\max}^2 \tau_{\min}(1-\gamma^{2H})}{2T(1-\gamma^2)}\left(\mathrm{KL}(\rho\|\mu) + \ln\frac{2}{\delta}\right)}$
   |  |  ```/* Reset prior and inject posterior knowledge to the actor   */```
19 |  |  $\mu \leftarrow \rho$                                     `// Reset prior to match posterior`
20 |  |  $\theta_{\mathrm{new}} \leftarrow \lambda \cdot \theta_{\mathrm{sampled}} + (1-\lambda) \cdot \theta_{\mathrm{current}}$
21 |  |  _load_policy_params$(\theta_{\mathrm{new}})$                       `// load into actor network`
22 |  |  $\lambda \leftarrow \lambda \cdot$ decay_rate     `// Ensure the actor converges to a stable policy`
23 |  |  clear_rollouts$(\mathfrak{D})$                         `// To start fresh in the next update`
24 |  **end**
25 **end**
26 **return** $\pi_\theta$, $\rho$, *and bound*

---

## D  Hyperparameter Selection

We carefully selected hyperparameters for our PAC-Bayes Soft Actor-Critic (PB-SAC) implementation to balance performance, sample efficiency, and theoretical guarantees. Our approach involves two sets of hyperparameters: those for the base SAC algorithm and those specifically for the PAC-Bayesian mechanisms.

### D.1  SAC Hyperparameters

For the base SAC algorithm, we used standard hyperparameters that have proven effective across continuous control tasks:

- Discount factor $\gamma = 0.99$
- Target network smoothing coefficient $\tau = 0.005$
- Batch size of 256 samples

- Learning starts after 5,000 environment steps
- Policy learning rate $\alpha_\pi = 3 \times 10^{-4}$
- Q-function learning rate $\alpha_Q = 1 \times 10^{-3}$
- Policy updates delayed by factor of 2 compared to critic updates
- Automatic entropy tuning enabled with initial temperature $\alpha = 0.2$

These parameters were chosen based on previous work by [18], with slight adjustments for our environments. The automatic entropy tuning is particularly important as it allows the algorithm to adapt the exploration-exploitation trade-off according to the complexity of the environment.

## D.2 PAC-Bayes Specific Hyperparameters

The PAC-Bayesian framework introduces several additional hyperparameters:

- PAC-Bayes update frequency of 20,000 environment steps
- KL regularization coefficient $\beta = 1.0$
- Posterior knowledge injection coefficient $\lambda = 0.01$
- Failure probability $\delta = 0.05$ (95% confidence level)
- Initial maximum reward estimate $R_{\max} = 1.0$
- 10,000 rollout trajectories for bound computation
- 75 steps per rollout trajectory

The infrequent PAC-Bayes updates (every 20,000 steps) are a critical design choice that balances computational efficiency with the need to maintain accurate performance guarantees. This allows the base SAC algorithm to make rapid progress between bound computations while ensuring the posterior distribution properly tracks policy improvements.

We deliberately set the posterior knowledge injection coefficient $\lambda$ to a small value (0.01) to ensure that the standard SAC optimization process dominates learning, while the PAC-Bayesian posterior provides a stabilizing influence and theoretical guarantees. This proved more effective than larger values, which tended to slow convergence by disrupting the actor's learning dynamics.

For bound computation, we found that 10,000 rollout trajectories of 75 steps each provides sufficiently accurate estimates of the mixing time and expected returns for our environments. These rollouts are performed with deterministic policies to accurately reflect the posterior's expected performance.

The PAC-Bayes updates maintain a posterior distribution over policy parameters, optimize it using a combination of critic values and KL regularization, and periodically inject information from this posterior into the actor network. This process allows us to derive high-probability bounds on policy performance without significantly hampering the learning capabilities of the base SAC algorithm.

