# OpenReview forum: "Towards Safe and Generalizable Treatment Strategies in Healthcare via RL and PAC-Bayesian Computations"
_NeurIPS.cc/2025/Conference — Submitted to NeurIPS 2025_

### Official Review · Reviewer_paMc · 2025-06-23

**Clarity:** 2
**Significance:** 2
**Originality:** 2
**Rating:** 3
**Confidence:** 2

**Summary:**

This work introduces a PAC-Bayesian framework for RL, which induces a novel soft actor critic based algorithm with guided exploration. Experiments are conducted to validate the theoretical results and the effectiveness of the proposed algorithm.

**Questions:**

1. The title kind of deviates the main content of this work. Some key words, like 'safe', 'generalizable' are not well defined and discussed in the main context. What is the formal definition of 'generalization', and how your methodology enhances the generalization ability? 'Safety' only appears several times in the manuscript without specific meaning. How does your method facilitate safety? How does it compare with other safe RL algorithms? For 'healthcare', it seems that the author only use application in healthcare as a motivating example, actually the method is designed for broader applications. Limited experiments (only sepsis) are conducted to evaluate the validity of the framework in healthcare. In sum, I suggest the authors considering revising the title, and provide formal and detailed discussion on 'safe' and 'generalization'.

2. As for the writing, too much space are allocated to preliminaries and background information from page 2 to page 5. I think we can go into the methodology part more earlier. Section 2 can be concised to leave space for more discussion and a formal presentation of the algorithm.

3. There is a line of work using sequential decision making for infectious disease control: 'Optimal treatment allocations in space and time for online control of an emerging infectious disease' and 'Deep spatial q-learning for infectious disease control'. Can you include them in discussion?

4. In line 43-44, 'Most standard generalization bounds and concentration inequalities rely on the assumption that samples are independent and identically distributed (i.i.d.)', to the best of my knowledge, concentrations for martingale are commonly used in theoretical RL works, such as 'Distributionally robust off-dynamics reinforcement learning: Provable efficiency with linear function approximation', 'Minimax optimal and computationally efficient algorithms for distributionally robust offline reinforcement learning'. I think the discussion can be revised here to focus on what is the theoretical contribution of this paper.


5. This paragraph is related to the Q1, in line 195-196, 'This PAC-Bayesian formalism allows us to reason about the generalization properties of randomized policies drawn from $\rho$'. Can you elaborate on this?

6. For theorem 3.2, equation (8),  how is the left hand side related to the prior $\mu$?

7. In line 267, 'we fit a posterior distribution $\rho$', can you elaborate why and how the $\rho$ is obtained like this, instead of from the lower bound in (9)?

8. For the experiment, current results do not show advantage over existing baselines. One could question the significance of the proposed method. Moreover, safety and generalization properties are not fully validated by the limited experiments. Seeds are not reported in the manuscript.


Minor issues:

1. equation (3), $n$ should be $m$;

2. equation (5), $=$ in the definition;

3. equation (6), I guess the event in the indicator should be $\neq$

4. Line 278, $H> R^4/(1-\gamma)^4$, H should be T?

**Ethical Concerns:**

["NO or VERY MINOR ethics concerns only"]

**Final Justification:**

Since most of my questions are of clarification type, the authors have provided detailed responses and addresses them properly. The paper is now clearer in terms of its methodology and contribution. I will raise my score by one point.

**Limitations:**

Yes.

**Quality:**

2

**Strengths And Weaknesses:**

The insight of operating the PAC-Bayes value-error bound in Theorem 3.2 for exploration and exploitation is interesting. However, the current manuscript has several questions to be addressed to justify its significance and contribution. Please see details below.

---

> ### Author Rebuttal · Authors · 2025-07-30
>
> # Common response to all reviewers
>
> We thank the reviewers for their time and effort in reviewing our work. We are glad that our work combining PAC-Bayes + RL + Healthcare setting has received several favourable comments, as expressed in "The application of the PAC-Bayesian framework to reinforcement learning is novel and insightful", "bound has improved constants relative to earlier work on PAC-Bayes bounds in RL", and "The insight of operating the PAC-Bayes value-error bound in Theorem 3.2 for exploration and exploitation is interesting".
>
> We also acknowledge the concerns and criticisms, addressed individually below.
> While the healthcare setting is part of the novelty in our contribution, we realize that there is a need to better contextualize our work, especially the claims regarding safety, to avoid unintended overclaiming/overselling and generally to improve the clarity. We intend to address this in the revised paper. For instance, to bring the title into a shape that represents the work better, we're considering "PAC-Bayesian Reinforcement Learning Trains Generalizable Treatment Policies" if this sits well. Likewise, we have also streamlined the abstract, and we'll do the corresponding updates in the revised paper to address your feedback.
>
> Should our paper be accepted, we commit to implementing the improvements described in our rebuttal, and to generally shape the paper into a high-quality contribution.
> We do believe our work contributes an interesting combination of PAC-Bayes, RL, and a healthcare setting demonstration, which deserves the attention of the NeurIPS community; and so we kindly ask the reviewers to reconsider their evaluation to support acceptance.
>
> # Response to Reviewer paMC
>
> Thank you for your thorough review and constructive feedback. We appreciate your careful reading and will address each of your concerns.
>
> ## AQ1, Regarding the title and definitions of "safe" and "generalizable":
>
> Very interesting point was raised and we thank the reviewer for this. Therefore, we should formalize these concepts:
>
> - **Generalization:** Our PAC-Bayes bound (Theorem 3.2) provides a high-probability guarantee that the deviation of the negative empirical return from its mean (the true expected return) is bounded as per the components of the upper bound. This directly quantifies generalization, how well a policy trained on training trajectories performs on new patient trajectories.
> - **Safety:** For a pre-set (user-chosen and typically problem-dependent) level $\delta$, our framework provides formal certificates that $\mathbb{P}(\texttt{return} \ge \texttt{critical\\_threshold}) > 1-\delta$ before deployment. For healthcare, this means verifying that treatment policies meet predefined safety standards (e.g., survival probability).
>
> We agree the title could be more precise. A revised title like the one in our common response above may better reflect the content.
>
> ## AQ2, Regarding writing and space allocation:
>
> We acknowledge that the extensive preliminaries (pages 2-5) do compress our methodological contributions. However, as noted by Reviewer yN38, it helps contextualize the contribution. In the revision, we will aim to find a better balance by condensing Section 2 to make more room for the algorithm presentation and theoretical insights.
> This is easily doable, we already have a good start on this and the final version will show these preliminaries compressed so that readers can reach the experiments sooner, as requested.
>
> ## AQ3,  Regarding infectious disease control literature:
>
> Thank you for these references. This line of work is indeed relevant, and we will include it in our discussion.
>
> ## AQ4, Regarding martingale concentration inequalities:
>
> Thank you for pointing out this imprecision. In lines 43-44, we were referring to standard generalization bounds outside the RL domain (hence citations [5,6] to Vapnik-Chervonenkis and Hoeffding). We mistakenly included citation [7] (Azuma) which indeed concerns martingales and should have appeared in our related work discussion where we already mention them. Easy to fix and this will be addressed in the final version.
>
> You are correct in noting that martingale-based concentration inequalities are well established in RL theory as your cited papers exemplify. Our contribution is distinct in different ways:  Rather than constructing or identifing martingale sequences from the RL data (e.g., value function errors, Bellman residuals), then applying martingale concentration inequalities (Azuma-Hoeffding, Freedman), we directly leverage the Markov chain structure, and apply a concentration inequality for Markov chains with explicit constants. What drives us is that RL problems often fail to naturally yield martingales, but they all have Markov structure by design. In any way, we'll revise this paragraph to clarify we meant general ML bounds require i.i.d. assumptions, while acknowledging the rich literature on martingale methods in RL in related works section.
>
> ## AQ5, Elaboration on generalization properties (line 195-196):
>
> The PAC-Bayesian formalism enables generalization reasoning through the KL divergence from the prior to the posterior. Specifically, the bound holds uniformly over all distributions $\rho$, in particular the posterior chosen after seeing data. The KL term penalizes posteriors that deviate from the prior, thus we obtain guarantees for an entire distribution of policies.
>
> ## AQ6, Relation between LHS and prior $\mu$ in Theorem 3.2:
>
> The expectation in LHS is over the posterior $\rho$. The prior $\mu$ appears on the RHS through $\mathrm{KL}(\rho \| \mu)$. This is a direct result of applying the change-of-measure inequality in Appendix B.5.2.
>
> ## AQ7, Regarding posterior fitting:
>
> Indeed, this requires clarification.  For obtaining $\rho$, we cannot directly optimize the lower bound in (9) because it's non-convex, there are two known solutions: **I.** construct an inverted KL version of it and optimize, but the inverse KL can only be estimated using bisection for example, and the method is sensitive to gradient explosion. **II.** construct a quazi-convex version (Thiemann et al. 2017, Masegosa et al. 2020) and use alternated optimization (Masegosa et al. 2020 Appendix H). Using this method, the RHS of Eq(9) becomes of the form
> $$-\underset{\theta \sim \rho}{\mathbb{E}}\left[\hat{\mathcal{L}}\_{\mathfrak{D}}(\theta)\right] - f(\lambda)\mathrm{KL}(\rho\Vert\mu)$$
> This optimization scheme can also be slow and impractical for RL (optimize $\lambda$ for a fixed $\rho$ and vice-versa). That's why we took a practical decision of absorbing $f(\lambda)$ into a scalar $\beta$ that intuitively tells how much of KL regularization we want to enforce.
> Finally, the term $-\underset{\theta \sim \rho}{\mathbb{E}}\left[\hat{\mathcal{L}}\_{\mathfrak{D}}(\theta)\right]$ is just approximated with a reliable empirical estimate taken from the critic network $\mathbb{E}\_{\theta \sim \rho}[-Q(s, \pi\_\theta(s))]$.
>
> ## AQ8, Experimental results and validation:
>
> We acknowledge that PB-SAC doesn't dramatically outperform baselines in terms of return. However, the significance of our work lies in providing both competitive performance and formal guarantees. (Please refer to our response to Reviewer zDTS regarding clinical relevance.)
>
> Regarding the seeds: we used 5-10 seeds (line 308)
>
> ## Minor Issues:
>
> Thank you for highlighting those typos, we corrected all of them. For the last one, That condition from prior works depend on the total number of observed samples (they denote it with $n$) and in our work, it should be $HT$ (or $N$ which is defined in line 185) instead of $H$.
>
> We hope this addresses all your concerns and clarifies our contributions. We greatly appreciate your detailed feedback and we remain open to further discussion if any points require additional clarification.

---

> > ### Comment · Reviewer_paMc · 2025-08-04
> >
> > Thank you for the detailed response, which addresses all my questions. The paper is now clearer in terms of its methodology and contribution. I encourage the authors to make further revisions to the manuscript to improve clarity based on the reviews. Accordingly, I will raise my score by one point.

---

> > > ### Author Response · Authors · 2025-08-06
> > >
> > > Thank you very much for raising your score and for your constructive engagement throughout this review process. We sincerely appreciate your recognition that our responses have clarified the methodology and contributions.
> > >
> > > We are committed to incorporating all the valuable feedback received during this review phase. Your insights, along with those from other reviewers, will help us deliver a clearer and more impactful final manuscript.
> > >
> > > Thank you again for your time and thorough evaluation of our work.

---

### Official Review · Reviewer_zDTS · 2025-06-27

**Clarity:** 3
**Significance:** 2
**Originality:** 2
**Rating:** 3
**Confidence:** 3

**Summary:**

proposes a PAC-bayesian bound on estimating the value of a policy in RL. By establishing a principled bound, this work can enable the deployment of RL in sensitive settings scuh as healthcare. to show that their bounds are non-vacuous in practice, the authors develop off-policy RL algorithms based on their bound and test in a sepsis simulator and some mujoco environments.

**Questions:**

The extension from IID to mixing is usually straightforward in learning theory and RL theory. What makes it challenging, if at all, in your setup?

Estimating T_mix seems like a hard problem. How will this be done in practice?

Deployed policies in real systems might have very high mixing time. What can be done in the real world? Why do you think your method will work outside of simulations, ie in the real world?

**Ethical Concerns:**

["NO or VERY MINOR ethics concerns only"]

**Final Justification:**

The author response to my question regarding technical innovation to handle mixing extension made me raise my score by 1 point.

**Limitations:**

The simulation to reality gap is most likely substantial here

**Quality:**

3

**Strengths And Weaknesses:**

Strengths:
- tackles in important problem. guarantees bounds can help accelerate deployment of RL in clinical settings
- bound has improved constants relative to earlier work on PAC-Bayes bounds in RL
- the authors don't just prove a bound but also derive and algorithm and test it in simulations

Weaknesses
- novelty seems limited. once you assume mixing, you're almost in IID setup. After T_mix samples, you basically get an almost independent sample. so the claims that they had to extend existing PAC Bayes theory to dependent RL setting are a little overstated. IID to mixing extensions are consider fairly straightforward.
- knowledge of T_mix is required. In experiments, the authors claim that they estimated T_mix from autocorrelations. This is not very convincing. Outside of simulations, I doubt whether T_mix can be reliably estimated from data.
- experiments are of doubtful clinical relevance beyond the sepsis simulator

---

> ### Author Rebuttal · Authors · 2025-07-30
>
> # Common response to all reviewers
>
> We thank the reviewers for their time and effort in reviewing our work. We are glad that our work combining PAC-Bayes + RL + Healthcare setting has received several favourable comments, as expressed in "The application of the PAC-Bayesian framework to reinforcement learning is novel and insightful", "bound has improved constants relative to earlier work on PAC-Bayes bounds in RL", and "The insight of operating the PAC-Bayes value-error bound in Theorem 3.2 for exploration and exploitation is interesting".
>
> We also acknowledge the concerns and criticisms, addressed individually below.
> While the healthcare setting is part of the novelty in our contribution, we realize that there is a need to better contextualize our work, especially the claims regarding safety, to avoid unintended overclaiming/overselling and generally to improve the clarity. We intend to address this in the revised paper. For instance, to bring the title into a shape that represents the work better, we're considering "PAC-Bayesian Reinforcement Learning Trains Generalizable Treatment Policies" if this sits well. Likewise, we have also streamlined the abstract, and we'll do the corresponding updates in the revised paper to address your feedback.
>
> Should our paper be accepted, we commit to implementing the improvements described in our rebuttal, and to generally shape the paper into a high-quality contribution.
> We do believe our work contributes an interesting combination of PAC-Bayes, RL, and a healthcare setting demonstration, which deserves the attention of the NeurIPS community; and so we kindly ask the reviewers to reconsider their evaluation to support acceptance.
>
> # Response to Reviewer zDTS
>
> Thank you for your thoughtful review. We appreciate your recognition of the importance of our problem and would like to address your concerns.
>
> ## Regarding novelty and the IID-to-mixing extension:
>
> While we appreciate the perspective, we respectfully note that the extension is not as immediate as it might appear. Although it is true that after $t_{mix}$ steps, the distribution of $X_{t_{mix}}$ is close to the stationary distribution $\nu$, however, it doesn’t guarantee that $X_{t_{mix}}$ is approximately independent of $X_0$, let alone consecutive states $X_{t_{mix}}$ and $X_{t_{mix}+1}$, those are definitely not independent. A broad vision can be found in the referenced work of Paulin (2018) where they introduce the technique of partitioning the chain (with $t_{mix}$ as partition size) and constructing a Marton coupling for the partition whose mixing matrix satisfies Eq(2.1) in Proposition 2.4 of Paulin (2018). From that we can infer that dependence decays exponentially with the mixing time, therefore, for approximate independence we usually need to wait longer, often several times $t_{mix}$.
>
> Regarding the second part of the question on what makes the theoretical development challenging. While we do not claim that our approach was uniquely difficult, we can nonetheless highlight several challenges we encountered during the development: First, to apply McDiarmid-type inequality for dependent random variables, we need to prove that our chosen function (negative empirical return) satisfies a bounded-differences condition with real and explicit (especially for RL) constants and this is what we've shown in Lemma 3.1 with its proof in Appendix (B.1, B.2, and B.3). Additionally, that derivation must carefully track how this perturbation propagates to future states, and this was discussed and proven in Appendix B.4. Another interesting discussion on the contrast to the literature of PAC-Bayes RL can be found in the paragraph at line 277 and Appendix B.6. Finally, The algorithmic part was as challenging as the theoretical part, the main difficulty lies in maintaining the statistical guarantee provided by the bound without hindering learning, because the KL penalizes straying too far from the prior. Thus, we proposed the prior resetting mechanism that helps controlling the KL, but we doubted it will violate the theoretical guarantees, that's why to make sure we respect them the algorithm collects fresh data at each update using a frozen policy network.
>
> ## Regarding $\tau_{min}$ estimation:
>
> We direct the reviewer to our response to Reviewer pTac addressing this issue.
>
> ## Regarding clinical relevance:
>
> We agree that simulation-to-reality gap is a concern, as stated in the title "**Towards** Safe and Generalizable...", which is why we view our work as providing one component of a comprehensive safety framework for clinical RL deployment, not a complete solution. More discussion can be found in the answer to question 2 of reviewer yN38, as well as the first part of our response to Reviewer pTac.

---

> > ### Comment · Reviewer_zDTS · 2025-08-01
> >
> > Thanks to the authors for explaining the challenges in the theoretical extension from iid to mixing setting. I am willing to increase my score by a point.

---

> > > ### Author Response · Authors · 2025-08-01
> > > **Thank you**
> > >
> > > We really appreciate this and we look forward to your revised score.
> > >
> > > Should you have any further thoughts on our work, please do not hesitate to post a comment for us and we'd be happy to respond as soon as possible during the current phase. Thanks for your time and consideration.

---

### Official Review · Reviewer_pTac · 2025-07-03

**Clarity:** 3
**Significance:** 3
**Originality:** 3
**Rating:** 4
**Confidence:** 3

**Summary:**

This paper proposes a novel PAC-Bayesian generalization bound for reinforcement learning (RL) in healthcare, addressing the critical challenge of temporal dependencies in Markovian data. The authors integrate bounded-differences conditions with McDiarmid-style concentration inequalities to derive a bound that explicitly depends on the Markov chain’s mixing time. They validate the framework through PB-SAC, an algorithm combining the bound with Soft Actor-Critic, demonstrating its effectiveness in simulated healthcare environments and continuous-control benchmarks. The work bridges theoretical RL guarantees with clinical applicability, providing high-probability confidence certificates for treatment policies.

**Questions:**

1. Does there exist typos in Eq.(6)?

See Weaknesses part.

**Ethical Concerns:**

["NO or VERY MINOR ethics concerns only"]

**Final Justification:**

After considering the authors' rebuttal and engaging in discussion, I am maintaining my recommendation for acceptance. The authors have satisfactorily addressed the initial concerns I raised regarding the paper's theoretical assumptions and the clarity of its presentation. Their responses were thorough and demonstrated a clear understanding of the limitations and future directions for their work.

The key issues have been effectively resolved. The authors acknowledged the limitations of the finite mixing time assumption and the use of KL divergence in high-dimensional spaces, framing them appropriately as avenues for future research.

**Limitations:**

yes

**Quality:**

3

**Strengths And Weaknesses:**

### Strengths
1. The paper proposes a novel PAC-Bayesian generalization bound combined with the mixing time of Markov chains to solve the problem of sequence data dependence in RL.
2. The effectiveness of proposed algorithm PB-SAC were verified in medical simulation and continuous control tasks.

### Weaknesses
1. The theory assumes the Markov chain has a finite mixing time, but real clinical data might not always meet this. The paper doesn't test how the bound performs when mixing time estimates are wrong or in non-stationary environments.
2. The authors note that using KL divergence between posteriors and priors can cause problems in high dimensions. It will be interesting to discuss alternative approaches.
3. The presentation of the experimental results is not very clear. Especially Figure 2(c) is confusing.

---

> ### Author Rebuttal · Authors · 2025-07-30
>
> # Common response to all reviewers
>
> We thank the reviewers for their time and effort in reviewing our work. We are glad that our work combining PAC-Bayes + RL + Healthcare setting has received several favourable comments, as expressed in "The application of the PAC-Bayesian framework to reinforcement learning is novel and insightful", "bound has improved constants relative to earlier work on PAC-Bayes bounds in RL", and "The insight of operating the PAC-Bayes value-error bound in Theorem 3.2 for exploration and exploitation is interesting".
>
> We also acknowledge the concerns and criticisms, addressed individually below.
> While the healthcare setting is part of the novelty in our contribution, we realize that there is a need to better contextualize our work, especially the claims regarding safety, to avoid unintended overclaiming/overselling and generally to improve the clarity. We intend to address this in the revised paper. For instance, to bring the title into a shape that represents the work better, we're considering "PAC-Bayesian Reinforcement Learning Trains Generalizable Treatment Policies" if this sits well. Likewise, we have also streamlined the abstract, and we'll do the corresponding updates in the revised paper to address your feedback.
>
> Should our paper be accepted, we commit to implementing the improvements described in our rebuttal, and to generally shape the paper into a high-quality contribution.
> We do believe our work contributes an interesting combination of PAC-Bayes, RL, and a healthcare setting demonstration, which deserves the attention of the NeurIPS community; and so we kindly ask the reviewers to reconsider their evaluation to support acceptance.
>
> # Response to Reviewer pTac
>
> Thank you for your constructive review and for recognizing the novelty of our approach. We appreciate your thoughtful questions and concerns.
>
> ## Regarding finite mixing time and robustness:
>
> You raise an excellent point about the mixing time assumption. Our theoretical bound indeed requires $\tau_{min}$ to be finite, and this is a clear limitation if we don't meet this need in practice. Alternative approaches could use spectral methods that can provide tighter bounds in addition, but they require estimating the spectral (or pseudo spectral) gap, which necessitate sufficient state-action coverage: We need enough visitation counts $N(s,a,s')$ to construct a reliable empirical transition matrix $\mathcal{\hat{P}}_n$. However when the state space is continuous, constructing transition matrices is infeasible.
>
> In practice (Algorithm 1, line 17), we use autocorrelation decay of the reward signal to estimate $\tau_{min}$ because it is simple and can be computed from streaming trajectories without storing full visitation counts. If an overestimation happens, our bound remains valid but becomes potentially looser, so it's harmless in practice. If someone wants a tight bound, they can resort to the option of collecting more trajectories, though it can also be costly as discussed in the answer to question 1 of reviewer yN38. However, on the other hand, underestimation can be a problem, because the bound becomes tighter causing overconfidence; one potential (untested) solution is to compute autocorrelation not only from the reward but from other information that flows in the chain (e.g., vital signs extracted from the states over time), this way we can verify from multiple sources.
>
> ## Regarding KL divergence alternatives:
>
> We completely agree about KL limitations in high dimensions. As mentioned in our limitations (Section 5), Wasserstein-based PAC-Bayes bounds might offer compelling advantage by respecting parameter space geometry. We view this along with avoiding posterior collapse as an exciting direction for future work that could significantly tighten theoretical certificates.
>
> ## Regarding Figure 2(c) clarity:
>
> We apologize for the confusion. Indeed Figure 2(c) isn't clear, it shows episodic returns for Humanoid, where PB-SAC (pink) closely tracks SAC (green) despite the additional PAC-Bayes regularization. The apparent visual complexity arises from the high variance in Humanoid and from not properly taking the mean over different runs. We will fix this issue as well as the small font size in the legend in the final version.
>
> ## Regarding Equation (6):
>
> Yes, the event in the indicator should be $\neq$.
> Thank you for noticing the typo! It's fixed now.

---

### Official Review · Reviewer_yN38 · 2025-07-03

**Clarity:** 3
**Significance:** 2
**Originality:** 3
**Rating:** 4
**Confidence:** 4

**Summary:**

This paper proposes a novel PAC-Bayesian generalization bound for reinforcement learning that explicitly accounts for the temporal dependencies in Markov Decision Processes—addressing a key limitation of existing generalization results, which are often too loose or assume independence. The authors introduce a bounded-differences condition on negative empirical return and apply a McDiarmid-style concentration inequality adapted to Markovian data, yielding a bound with explicit dependence on the mixing time of the underlying Markov chain. They demonstrate its applicability to off-policy RL algorithms like Soft Actor-Critic and show that it provides meaningful confidence guarantees in simulated healthcare settings. Overall, the work offers a theoretically grounded and practical framework for deploying safe and reliable RL-based treatment policies in high-stakes domains like personalized medicine.

**Questions:**

1. The title emphasizes healthcare, and the paper notes that a key challenge in building trust in RL for healthcare applications is the sequential and dependent nature of RL data. However, such data dependencies are common in many real-world domains and not unique to healthcare. Could the authors elaborate on what makes healthcare particularly special in this context? Providing a more detailed framework or concrete examples from healthcare might better justify the title and clarify the domain-specific significance.

2. The theoretical development and explanation of PAC-Bayesian RL remain quite abstract and high-level. It’s difficult to see how this theory directly connects to practical applications in healthcare. Could the authors provide specific, grounded examples that illustrate the advantages of using PAC-Bayes RL in healthcare settings, rather than relying solely on general PAC learning bounds? This would help clarify both the motivation and the practical value of the proposed framework.

**Ethical Concerns:**

["NO or VERY MINOR ethics concerns only"]

**Final Justification:**

I suggested that the authors formulate the problem as a new healthcare MDP. They were responsive and ultimately proposed a solid framework, though I believe it would benefit from further feedback from other reviewers. I am willing to recommend a borderline accept at this stage.

**Limitations:**

yes

**Quality:**

3

**Strengths And Weaknesses:**

Strength:
The application of the PAC-Bayesian framework to reinforcement learning is novel and insightful. The background on PAC learning is well-introduced and helps contextualize the contribution.

Weakness:
While the paper proposes a new framework for safe and generalizable treatment strategies in healthcare using PAC-Bayesian reinforcement learning, the motivation could be more compelling. The comparison between traditional RL and PAC-Bayes RL, as well as between standard PAC analysis and PAC-Bayes, is not sufficiently discussed. Additionally, the theoretical results remain high-level and lack concrete guidance or actionable insights for real-world healthcare applications.

---

> ### Author Rebuttal · Authors · 2025-07-30
>
> # Common response to all reviewers
>
> We thank the reviewers for their time and effort in reviewing our work. We are glad that our work combining PAC-Bayes + RL + Healthcare setting has received several favourable comments, as expressed in "The application of the PAC-Bayesian framework to reinforcement learning is novel and insightful", "bound has improved constants relative to earlier work on PAC-Bayes bounds in RL", and "The insight of operating the PAC-Bayes value-error bound in Theorem 3.2 for exploration and exploitation is interesting".
>
> We also acknowledge the concerns and criticisms, addressed individually below.
> While the healthcare setting is part of the novelty in our contribution, we realize that there is a need to better contextualize our work, especially the claims regarding safety, to avoid unintended overclaiming/overselling and generally to improve the clarity. We intend to address this in the revised paper. For instance, to bring the title into a shape that represents the work better, we're considering "PAC-Bayesian Reinforcement Learning Trains Generalizable Treatment Policies" if this sits well. Likewise, we have also streamlined the abstract, and we'll do the corresponding updates in the revised paper to address your feedback.
>
> Should our paper be accepted, we commit to implementing the improvements described in our rebuttal, and to generally shape the paper into a high-quality contribution.
> We do believe our work contributes an interesting combination of PAC-Bayes, RL, and a healthcare setting demonstration, which deserves the attention of the NeurIPS community; and so we kindly ask the reviewers to reconsider their evaluation to support acceptance.
>
> # Response to Reviewer yN38
>
> Thank you for your thoughtful review. We appreciate your recognition of the novelty and insight in applying the PAC-Bayesian framework to reinforcement learning.
>
> ## AQ1, Regarding healthcare-specific significance:
>
> The healthcare setting presents unique challenges beyond other decision-making domains, we try to outline how our work meets some of those challenges:
>
> - **Irreversibility of failures.** Unlike other domains where the consequences of failures can be damaged equipment (robotics) or wasted compute (LLMs), failures in medical decisions can cause permanent harm. A statistical guarantee can certify an RL algorithm and enable clinicians to verify that $\mathbb{P}(\texttt{return} \ge \texttt{critical\\_threshold}) > 1-\delta$ (i.e. with high probability, the return won't drop bellow threshold) before deployment.
> - **Regulatory requirements.** Healthcare algorithms face strict approval processes requiring formal safety guarantees. Theoretical certificates of the kind we put forward in this work are a step towards addressing this need.
> - **Limited exploration.** One cannot freely explore treatment options on patients. One may infer that our bound in Eq. 9, with UCB-like structure, helps guide safe exploration (the KL penalizes going too far from the prior), which is beneficial for off-policy learning from historical data.
>
> To address your concern comprehensively, we will add the following paragraph to our revised paper to clarify these domain-specific challenges:
> "Healthcare presents distinct challenges for reinforcement learning that go beyond those encountered in domains such as robotics or chemistry. First, the heterogeneity of environments and variables is particularly pronounced: medical datasets often contain dozens of interdependent variables (vital signs, lab results, medication history), and even after feature selection, discarding variables risks losing essential clinical information. Second, partial observability is intrinsic to medical data, despite all diagnostic efforts, the true physiological state remains fundamentally uncertain, necessitating algorithms that can generalize robustly. Finally, exploration through online learning is ethically and practically infeasible. As a result, learning must be conducted offline using historical data, making our PAC-Bayes framework with its formal guarantees particularly valuable."
>
> ## AQ2, Regarding concrete healthcare guidance:
>
> As example to illustrate the advantage of using PAC-Bayes RL in a healthcare setting, consider the ICU-Sepsis experiments (Section 4). This experiment demonstrates practical value and addresses the request in the following sense. The bound certifies with $95$\% confidence ($\delta = 0.05$) that the survival probability exceeds a threshold, directly informing deployment decisions. For instance, in adaptive dosing for chronic conditions, our framework can guarantee that expected patient health scores won't fall below certain values, it only remains to verify whether those values align with the critical threshold defined by healthcare professionals, something where traditional RL cannot provide.
>
> As for the second part of the question about the benefit compared to general PAC learning bounds, we can identify two main things: **I.** General PAC learning bounds are heavily studied and provide rigorous guarantees but they are generally based on assumptions (e.g. independent data) not met in RL, and furthermore can be loose in many problems of interest. **II.** PAC-Bayes uniquely allows encoding prior clinical knowledge through $\mu$ (a.k.a prior-informed PAC-Bayes). We can for instance initialize the prior using existing clinical guidelines (e.g., surviving sepsis campaign protocols). This makes $\mathrm{KL}(\rho \| \mu)$ small when the learned policy aligns with established practices. Therefore tightening the bound compared to general PAC analysis that treats all policies equally.
>
> Thank you for highlighting these important aspects. We hope this clarifies how our work specifically addresses healthcare challenges while maintaining the theoretical rigor.

---

> > ### Comment · Reviewer_yN38 · 2025-08-04
> >
> > Thank you for your comprehensive rebuttal—I’ve carefully reviewed your responses. While I appreciate the clarity you provided throughout, I still believe that positioning healthcare as the central motivation will strengthen the overall story. In the next version, consider opening with real‑world medical scenarios to firmly ground the reader, then transition into a healthcare‑specific reinforcement learning formulation with all terminology clearly defined—possibly introducing a new concept or notation that’s especially relevant to the medical context—rather than first diving into general RL methods, PAC‑Bayes learning, or background material. Once that domain‑tailored setup is in place, you can delve into your technical solution. This reorganization—motivation → healthcare‑tailored setup → solution—will help readers immediately see why the problem matters and how your contributions directly map to the real‑world context.

---

> ### Author Response · Authors · 2025-08-06
>
> We took your suggestion very seriously and have restructured our introduction accordingly. The new structure opens with a concrete clinical scenario, establishes healthcare-specific challenges that motivate our work, and only then introduces the technical framework. Below is an excerpt:
>
> > ## Introduction
>
> > Sequential decision-making under uncertainty is particularly critical in healthcare settings. Consider sepsis treatment in the Intensive Care Unit (ICU), where clinicians face a cascade of interconnected decisions: when to initiate antibiotics, how to titrate intravenous fluids while monitoring organ function, whether to start vasopressor agents and at what dose, and if mechanical ventilation becomes necessary [1 ]. Each intervention shapes not only immediate physiological responses—blood pressure, oxygen saturation, urine output—but also influences which treatment options remain viable hours later. A fluid bolus that stabilizes blood pressure now might worsen pulmonary edema later; aggressive vasopressor dosing could restore perfusion but risk peripheral ischemia. These decisions unfold against a backdrop of incomplete information and patient heterogeneity, where two patients with similar presentations may respond differently to identical interventions. This exemplifies a broader challenge in healthcare: optimizing treatment sequences where each action constrains future possibilities, outcomes manifest over extended timescales, and the stakes of suboptimal decisions are measured in permanent organ damage or mortality rather than mere inefficiency.
>
> > Reinforcement learning (RL) offers a natural framework for such sequential decision-making, learning policies that adapt based on patient responses over time. Recent work has explored RL for critical care management [2 , 3 ], precision drug dosing [3], infectious disease control [x, x], and identifying treatment paths that avoid adverse outcomes [4 , 2 ]. These applications demonstrate RL’s potential to enhance clinical decision-making by learning from historical treatment data.
>
> > The healthcare setting imposes three fundamental constraints that distinguish it from general RL. **(1) Irreversibility of failures**: Unlike in other domains, where failures may result in damaged equipment (e.g., robotics) or wasted computational resources (e.g., LLMs), failures in medical decision-making can lead to permanent harm. Imprecisely calibrated therapeutic interventions, although potentially beneficial in addressing specific physiological deficits, can inadvertently worsen chronic conditions by disrupting compensatory mechanisms or exacerbating the patient’s overall state [ 5 ]. Thus, we require to accept only policies that highly deliver treatment outcomes above a minimum acceptable level *{footnote: For a pre-set (user-chosen and typically problem-dependent) confidence level $\delta$, We only accept policies that satisfy $\mathbb{P}(\texttt{return} \ge \texttt{critical\\_threshold}) > 1-\delta$ (i.e., with high probability, the return of such policy won't drop bellow threshold)}* before being used in real clinical settings. **(2) Regulatory requirements**: Healthcare algorithms face strict approval processes requiring formal safety guarantees [ 6 ], theoretical certificates of this kind are a step toward addressing this need. **(3) Limited exploration**: Ethical and practical constraints prohibit exploratory actions on patients. Learning must rely primarily on historical data, necessitating methods that can provide guarantees while guiding any limited exploration safely.
>
> > There is a widely unmet demand for rigorous, high-confidence guarantees on the generalization capabilities of machine learning models [ 7]. Providing such guarantees faces a fundamental theoretical challenge: the sequential, dependent nature of RL data violates the independence assumptions underlying most generalization results. Standard bounds in machine learning [8] and concentration inequalities [9 ] assume i.i.d. samples—clearly inappropriate for trajectories where each state depends on previous actions. While there has been developments on martingale-based concentration inequalities [10, 11] by constructing martingale sequences from value function errors or Bellman residuals, these require identifying suitable martingale structures that may not naturally arise in all problems. The PAC-Bayesian framework [12, 13 , 14, 15 ] offers an alternative path: it can provide data-dependent bounds while incorporating prior knowledge (valuable for encoding clinical guidelines). However, extending PAC-Bayes to handle Markovian dependencies remains relatively unexplored.
>
> > [Rest of introduction continues with some related works, contributions, and paper organization...]
>
> ---
>
> Thanks again for this valuable guidance. We believe it has significantly improved the paper's narrative flow. We would also appreciate your thoughts on our proposed revised title in the common response, and any further comments.

---

> > ### Comment · Reviewer_yN38 · 2025-08-07
> >
> > Thanks for revising the introduction — it now reads much more clearly and feels more reasonable to me. The three fundamental constraints you’ve identified that distinguish this setting from general RL are particularly helpful.
> >
> > Given this updated framing, could you please reintroduce the problem setup? Specifically, I’d like to see: (1) how this setup differs from traditional RL, (2) what your core objective is, and (3) how the setup formally reflects the three constraints mentioned in the introduction. At the moment, it feels like you’ve combined two basic setups without introducing much novelty.
> >
> > If you can present a clear and well-motivated setup, I’d be happy to raise my score to 4. Furthermore, if you can provide a non-trivial solution — ideally with a meaningful bound or theoretical insight that connects directly to healthcare implications (the current bound feels too generic to be informative) — I would consider raising my score to 5 and strongly support the paper.
> >
> > I understand this may be difficult to achieve in a short time frame, so if it’s not feasible now, I encourage you to consider refining the paper further and submitting it to the next conference.

---

> ### Author Response · Authors · 2025-08-07
>
> Thank you for the valuable comments and pushing us to develop a more meaningful healthcare-specific contribution.
>
> ## Our Response
>
> Your comment about needing a "non-trivial solution with healthcare implications" resonated deeply. Initially, we had been thinking about clinically reinterpreting standard PAC-Bayes bounds, but as you said these felt too generic.
>
> We therefore carefully rethought the notion of **irreversibility**. In healthcare, certain states represent permanent, irreversible harm. Which isn't just about negative rewards; but rather about states from which recovery is impossible.
>
> This realization reminded us of a very recent work by Richens et al. (2025) (and references therein) on the necessity of world models. They formalize the problem with controlled Markov processes (cMPs), which elegantly handles goal-directed behavior through temporal logic. Their framework defines goals in terms of reaching desirable states. However, we observed that, in healthcare, it is equally important to avoid *undesirable* states. This, led us to extend their framework for healthcare to explicitly model states that must never be entered.
>
> ## Problem Formulation
>
> We extend the standard RL formulation to a **healthcare MDP** $(S, A, P, R, \gamma, \mathcal{C})$, where $\mathcal{C} \subseteq S$ represents **irreversible critical states** (e.g., organ failure, death). This directly reflects our first constraint: the irreversibility of failures. We recognized that trajectories entering $\mathcal{C}$ are fundamentally distinct—they represent catastrophic failures with typically bounded, negative future returns.
>
> This motivated a natural decomposition of the value function: $$V^\pi = V^\pi\_{\text{safe}} \cdot (1 - \rho\_\mathcal{C}(\pi)) + V^\pi\_{\text{unsafe}} \cdot \rho\_\mathcal{C}(\pi)$$
>
> where $\rho\_\mathcal{C}(\pi) = \mathbb{P}(\text{entering } \mathcal{C})$ is the irreversibility risk.
>
> ## Developing the Bound
>
> The challenge was to apply PAC-Bayes to this decomposition. We could not treat all trajectories uniformly. We therefore:
>
> 1. Apply PAC-Bayes only to safe trajectories (those avoiding $\mathcal{C}$)
> 2. Separately bound the loss from unsafe trajectories
> 3. Combine them using the empirical frequency of entering $\mathcal{C}$
>
> This yields our main result that has been rigorously proven with the new setup—a bound with three interpretable components:
>
> $$V^\pi \geq \underbrace{\hat{V}^{\pi,\text{safe}}\_D}\_{\text{safe trajectory value}} - \underbrace{\sqrt{\frac{R^2\_{\max}\tau\_{\min}(1-\gamma^{2H})}{2T\_{\text{safe}}(1-\gamma^2)}\left(\text{KL}(\rho||\mu) + \ln\frac{2}{\delta}\right)}}\_{\text{PAC-Bayes penalty}} - \underbrace{\rho\_\mathcal{C}(\pi) \cdot \kappa}\_{\text{irreversibility penalty}}$$
>
> This bound has a direct clinical interpretability:
>
> - **$\hat{V}^{\pi,\text{safe}}\_D$**: Performance when treatment succeeds
> - $T\_{\text{safe}} = T \cdot (1-\rho\_\mathcal{C}(\pi))$ is the effective number of safe trajectories
> - $\kappa = V\_{\max} - V^{\pi}\_{\text{unsafe}}$: value loss from entering critical states (a pessimistic bound)
> - $\hat{\rho}\_\mathcal{C}(\pi) = \frac{T\_{\text{unsafe}}}{T} = \frac{T - T\_{\text{safe}}}{T}$: we can use this approximation empirically
>
> ## Addressing The Three Constraints
>
> Looking back, this bound naturally addresses all three key constraints in healthcare:
>
> **Addressing Irreversibility**: The decomposition into safe/unsafe trajectories with explicit penalty $\kappa$ quantifies the cost of irreversible failures. Unlike standard RL bounds that treat all trajectories equally, we separately bound the catastrophic loss from entering $\mathcal{C}$.
>
> **Regulatory Requirements**: This is indeed a mission-critical objective currently hampered by regulatory frameworks lagging behind AI advances and the scarcity of teams combining ML theory, clinical expertise, and regulatory knowledge.
>
> The bound provides a certificate: "With confidence $1-\delta$, the policy achieves a lower value bound while maintaining $\mathbb{P}(\text{adverse events}) \leq \rho\_\mathcal{C}(\pi)$."
>
> **Limited Exploration**: The dependence on $T\_{\text{safe}}$ shows we need sufficient data from trajectories that successfully avoided critical states.
>
> ---
> The key was recognizing that healthcare isn't just about maximizing expected returns, it's about doing so while providing guarantees against catastrophic, irreversible failures. By extending the cMP framework with explicit modeling of undesirable states, and formalizing their avoidance through Linear Temporal Logic (LTL), we derived this bound quickly in response to your comments, and we will implement experiments based on it in the final version.
>
> We believe this directly addresses both of your requests: **I.** Reintroducing the problem setup, and **II.** Providing a non-trivial bound or theoretical insight connected to healthcare implications.
>
> We’re grateful for your thoughtful and constructive feedback and for your openness to revise the score.

---

> > ### Comment · Reviewer_yN38 · 2025-08-08
> >
> > Thank you for your thorough and proactive response. I now find the proposed healthcare MDP framework to be much more helpful and have updated my score to borderline accept. Please revise the paper accordingly, and consider providing more interpretation and explanation of the results derived from this new framework. I will continue the discussion with the AC and other reviewers to gather their perspectives as well.

---

### Note · Authors · 2025-08-13

We sincerely thank all reviewers for their constructive engagement throughout this discussion period. The dialogue has been exceptionally productive, leading to substantial improvements in our work.

We are particularly grateful to Reviewer yN38 for pushing us to develop a more meaningful healthcare-specific framework. In response to their challenge, we developed a novel healthcare MDP formulation that explicitly models irreversible critical states. This led to our new PAC-Bayes bound that decomposes into safe/unsafe trajectories with direct clinical interpretability. We have added this healthcare-specific bound to the manuscript alongside our original general bound, as both provide complementary insights.

We also appreciate reviewer paMc's thorough examination of our technical contributions and their recognition of our clarifications regarding martingale concentration inequalities versus our Markov chain approach. Reviewer pTac's insights on mixing time estimation helped elaborate more on this technical aspect. Reviewer zDTS's questions about the IID-to-mixing extension helped us better articulate the theoretical challenges we addressed.

Following reviewer feedback, we have committed to several key improvements for the final version:
1. Restructured introduction opening with concrete clinical scenarios; developed and introduced the new healthcare MDP framework with its setup (per yN38)
2. Formalized definitions; Compressed Section 2 preliminaries; clarified our technical contribution versus martingale-based approaches; (per paMc)
3. Clearer experimental visualizations and mixing time discussion (per pTac)
4. Elaborated on theoretical challenges of IID-to-mixing extension (per zDTS)
5. Implemented the new healthcare MDP framework, and experiments are currently in progress

The reviewers' recognition of our work's novelty—combining PAC-Bayes, RL, and healthcare applications—along with the score improvements during discussion (paMc and zDTS raising scores, yN38 moving to borderline accept) demonstrates the value of this review process. We believe our revised work makes meaningful contributions to both safe RL deployment in critical domains and learning theory in RL, and would be honored if it were selected for acceptance. We remain committed to delivering a high-quality final manuscript that incorporates all feedback received.

Thank you again for your time, expertise, and constructive criticism that have substantially strengthened our contribution.

---

### Decision · Program_Chairs · 2025-09-17

**Decision:**

Reject

**Comment:**

The paper develops a PAC-Bayesian generalization bound for RL that explicitly accounts for temporal dependence via the mixing time of the underlying Markov chain. The bound is instantiated in an off-policy algorithm (PB-SAC) and used to produce high-probability performance certificates in a sepsis simulator and standard control tasks.

During discussion, the authors further proposed a healthcare-specific MDP with irreversible critical states and derived a decomposed PAC-Bayes bound that separates safe and unsafe trajectories, giving components with direct clinical interpretation (e.g., irreversibility risk and safe-trajectory performance).

We discussed a lot and actively interacting with authors for throughly evaluating the strengths and weaknesses of the work. The strengths of this paper are impressive. First, it shows theoretical novelty with practical impacts, with a value bound adapted to Markovian dependence and improved constants compared to prior RL PAC-Bayes results. The discussion clarified why the IID to mixing extension is nontrivial in this setting. The added irreversible-states healthcare MDP and the safe/unsafe decomposition make the guarantees interpretable to clinicians. This directly addresses reviewer requests for domain-specific grounding. Also, the proposed PB-SAC uses the bound for exploration–exploitation, with well design (e.g., prior resetting to control KL while preserving guarantees) and competitive returns alongside non-vacuous certificates.

The weaknesses of this work include: under some important settings (mixing time estimates are wrong and non-stationary settings), the effectiveness of proposed method remains to be demonstrated. The authors frankly agreed with that. Also, clinical evidence is limited to a single sepsis simulator; real-world datasets or broader medical environments are not yet included. Moreover, in terms of the current version, we concerned about the over-claiming. The title/abstract originally leaned into “safe”/“generalizable” without precise definitions. Review discussion improved this; the next version should reflect those changes.

Since the submissions cannot be updated during reviewing process, to significantly improve the clarification and impacts of the manuscript, the authors committed to update their paper on introduction, definition, experiments, and discussion, as in their final remarks. A potential risk is the change would be significant and leading to a new draft.

Taken together, the paper advances the interface between learning theory and deployable RL in healthcare. With these revisions as mentioned above (detailed in final justifications of reviewers and final remarks by the authors), the work will offer both more solid theory and actionable guidance for deploying RL with credible guarantees in sensitive domains. With the great potential merit of this work, we highly encourage the authors to resubmit to the next venue.